# Glycolytic flux controls retinal progenitor cell differentiation via regulating Wnt signaling

Joseph Hanna[1,2,3], Yacine Touahri[1,3,4], Alissa Pak[1,2], Lauren Belfiore[1,2], Edwin van Oosten[1,4], Luke Ajay David[1,2,3], Sisu Han[5], Yaroslav Ilnytskyy[6], Igor Kovalchuk[6], Deborah Kurrasch[5], Satoshi Okawa[7,8,9,10], Antonio del Sol[10,11,12], Robert A Screaton[1,4], Isabelle Aubert[1,2], Carol Schuurmans[1,2,3,4]*

[1]Biological Sciences, Sunnybrook Research Institute, Toronto, Canada; [2]Department of Laboratory Medicine and Pathobiology, University of Toronto, Toronto, Canada; [3]Department of Ophthalmology and Vision Sciences, University of Toronto, Toronto, Canada; [4]Department of Biochemistry, University of Toronto, Toronto, Canada; [5]Department of Medical Genetics, Hotchkiss Brain Institute, Alberta Children's Hospital Research Institute, University of Calgary, Calgary, Canada; [6]Department of Biological Sciences, University of Lethbridge, Lethbridge, Canada; [7]Pittsburgh Heart, Lung, and Blood Vascular Medicine Institute, University of Pittsburgh School of Medicine, Pittsburgh, United States; [8]Department of Computational and Systems Biology, University of Pittsburgh School of Medicine, Pittsburgh, United States; [9]McGowan Institute for Regenerative Medicine, University of Pittsburgh School of Medicine, Pittsburgh, United States; [10]Computational Biology Group, Luxembourg Centre for Systems Biomedicine, U of Luxembourg, Esch-sur-Alzette, Luxembourg; [11]CIC bioGUNE, Bizkaia Technology Park, Derio, Spain; [12]IKERBASQUE, Basque Foundation for Science, Bilbao, Spain

*For correspondence:
cschuurm@sri.utoronto.ca

## eLife Assessment

This **fundamental** study advances our understanding of the role that energy metabolism, specifically anaerobic glycolysis, plays during retinal development. **Convincing** in vitro genetic and pharmacological evidence demonstrates that glycolytic flux controls retinal progenitor cell proliferation rates and the timing of photoreceptor maturation. Interesting evidence suggests potential downstream roles for intracellular pH and Wnt/β-catenin signaling; however, more direct evidence is needed to show they are the key mechanisms through which glycolytic flux regulates retinogenesis in vivo. This work is expected to stimulate broad interest and possible future studies investigating the link between metabolism and development in other tissue systems.
[Editors' note: Primary data for this manuscript are not available due to a corrupted hard drive that occurred during the course of peer review. However, preprocessed data are available.]

**Abstract** Metabolic pathways are remodeled in response to energy and other homeostatic demands and are dynamically regulated during embryonic development, suggesting a role in guiding cellular differentiation. Here, we show that glycolytic flux is required and sufficient to bias multipotent retinal progenitor cells (RPCs) to acquire a rod photoreceptor fate in the murine retina. In RPC-specific *Phosphatase and tensin homolog* conditional knockout (*Pten*-cKO) and RPC-specific conditional gain-of-function of dominant active PFKFB3 (cytoPFKFB3) mice, glycolytic gene

expression and activity are elevated, correlating with precocious rod photoreceptor differentiation and outer segment (OS) maturation. Conversely, glycolytic inhibition in retinal explants suppresses RPC proliferation and photoreceptor differentiation, achieved either with 2-deoxy-D-glucose, a competitive inhibitor of glucose metabolism, by lowering media pH, which disables PKM2, a rate-limiting enzyme, or by inhibiting lactate/$H^+$ symporters, which lowers intracellular pH. Mechanistically, we show that Wnt signaling, the top-upregulated pathway in *Pten*-cKO retinas, is a glycolysis-dependent pathway. Pharmacological and genetic perturbation of Wnt signaling by knocking-out *Ctnnb1*, encoding β-catenin, phenocopies glycolytic inhibition, suppressing RPC proliferation, photoreceptor differentiation, and OS maturation. Thus, developmental rewiring of glycolytic flux modulates Wnt signaling to drive rod photoreceptor differentiation and maturation, an instructive role that may be exploited therapeutically for cell replacement strategies.

## Introduction

Multiple neural cell types comprise the mature retina, all of which must be produced in correct numbers with precise identities for effective visual processing. Included are rod and cone photoreceptors, the light-sensing neurons; amacrine, bipolar, and horizontal cells, retinal interneurons; ganglion cells, the output neurons of the retina; and supportive glial cells known as Müller glia. Clonal analyses across vertebrate species have revealed that all seven retinal cell types are derived from multipotent retinal progenitor cells (RPCs) that give rise to clones of varying sizes and compositions (*Alexiades and Cepko, 1997*; *Fekete et al., 1994*; *Gomes et al., 2011*; *Holt et al., 1988*; *Jensen and Raff, 1997*; *Turner and Cepko, 1987*; *Turner et al., 1990*; *Wetts and Fraser, 1988*). In mice, RPCs begin to differentiate at embryonic day (E) 12, completing their proliferation and differentiation by postnatal day (P) 10–12 (*Cepko, 2014*; *Young, 1985*; *Zhang et al., 2023a*). Symmetric proliferative (P/P) divisions predominate early in development, followed by a shift to asymmetric (P/D) divisions that renew the RPC pool and generate differentiated cells, and finally, by symmetric differentiative (D/D) divisions that generate two postmitotic cells (*Baye and Link, 2007*; *Nerli et al., 2020*; *Paolini et al., 2015*). At each stage, individual RPC fate decisions are largely stochastic, albeit with temporal biases, such that the seven retinal cell types are born in overlapping windows (*Gomes et al., 2011*; *He et al., 2012*). Ganglion, horizontal, and cone cells are born in the earliest embryonic stages, followed closely by amacrine cells, and finally by bipolar cells, rods, and Müller glia, which mainly differentiate in the first two postnatal weeks (*Rapaport et al., 2004*; *Young, 1985*). How RPCs interpret extrinsic signals and intrinsic contextual cues to ultimately produce the correct numbers of each retinal cell type remains incompletely understood.

Cellular energy and precursor metabolites are produced by the metabolism of glucose, which is converted into pyruvate via glycolysis. Glycolysis is a multi-step enzymatic process that occurs in the cytosol, does not require $O_2$, and produces 2 molecules of adenosine triphosphate (ATP) per glucose molecule (*Hanna et al., 2022*). Pyruvate is either transported to mitochondria for further oxidation to drive oxidative phosphorylation (OXPHOS) by the electron transport chain, or converted to lactate and secreted out of the cell. OXPHOS is an $O_2$-dependent metabolic pathway that produces 30–38 ATP molecules for each glucose molecule, and hence, produces more cellular energy than glycolysis (*Hanna et al., 2022*). Yet, the majority of vertebrate retinal cells rely on glycolysis as the main metabolic pathway, even in the presence of $O_2$ (*Hurley et al., 2015*). The preference for aerobic glycolysis has been termed the Warburg effect, an adaptation commonly observed in cancer cells that is also used by some 'normal' cells (*Vander Heiden et al., 2009*; *Warburg, 1956a*; *Warburg, 1956b*). Included are dividing neural progenitor cells, which use aerobic glycolysis for energy production across tissue types, including in the retina (*Hanna et al., 2022*). In *Xenopus*, RPCs rely on glycogen stores as a fuel source for glycolysis (*Agathocleous et al., 2012*), whereas in human retinal organoids, glucose is the essential fuel and is required for RPC expansion at the earliest stages of optic vesicle formation (*Takata et al., 2023*). Finally, during murine retinal development, mitophagy clears mitochondria to drive a metabolic shift toward glycolysis, which supports early-RPC differentiation into ganglion cells (*Esteban-Martínez et al., 2017*). However, whether metabolic reprogramming and levels of glycolytic activity influence RPC differentiation decisions during later developmental stages is not known.

Metabolic reprogramming, or alterations in metabolic rates and/or pathways, occurs in stem and progenitor cells during tissue development (*Tatapudy et al., 2017*), including in the retina

(*Esteban-Martínez et al., 2017*). In a few lineages, metabolism has been ascribed instructive roles in driving cellular differentiation, including in skeletal muscle and helper T cell lineages (*Peng et al., 2016*; *Ryall et al., 2015*; *Tatapudy et al., 2017*). In these lineages, glycolysis is important for more than energy production, for instance, modifying histone acetylation in murine skeletal muscle (*Ryall et al., 2015*), activating Wnt signaling to control chick neuromesodermal progenitor cell fate choice (*Oginuma et al., 2020*), and altering protein subcellular localization in the murine presomitic meso-derm (*Miyazawa et al., 2022*). To further investigate how developmental signaling pathways intersect with metabolism to control RPC fate choice in the retina, we focused on PTEN, an intracellular protein and lipid phosphatase with pleiotropic actions, including the direct inhibition of glycolysis through dephosphorylation and consequent inactivation of phosphoglycerate kinase 1 (PGK1), a rate-limiting glycolytic enzyme (*Qian et al., 2019*). In *Pten* conditional knockout (KO) retinas, RPCs proliferate excessively early in embryonic development, accompanied by the accelerated differentiation of photo-receptors in the outer nuclear layer (ONL) (*Cantrup et al., 2012*; *Jo et al., 2012*; *Tachibana et al., 2016*; *Tachibana et al., 2018*). Here, we show that elevated glycolysis accompanies the accelerated RPC differentiation kinetics in *Pten*-cKO retinas, which ultimately results in the precocious depletion of the RPC pool. Similarly, in an RPC-specific, conditional gain-of-function (GOF) model of domi-nant active PFKFB3 (cytoPFKFB3), glycolytic gene expression and activity are elevated, correlating with precocious rod photoreceptor differentiation and outer segment (OS) maturation. Conversely, multiple inhibitors of glycolysis reduce RPC proliferation and photoreceptor differentiation. Finally, we found that glycolysis is required and sufficient to control Wnt signaling, and show that Wnt activity is required for rod photoreceptor differentiation and OS maturation. We have thus uncovered a novel glycolytic flux–Wnt signaling axis as an important regulator of RPC fate choice.

## Results

### Elevated glycolysis and accelerated rod photoreceptor differentiation correlate in *Pten*-cKO retinas

Neural progenitor cells in the embryonic brain (*Khacho and Slack, 2018*) and retina (*Agathocleous et al., 2012*) preferentially use glycolysis for energy production. However, glycolytic metabolites and glycolytic gene expression are not at uniform levels throughout retinal development, peaking at E18.5–P0 (*Esteban-Martínez et al., 2017*). Since these measurements were performed in bulk, and the ratio of RPCs to differentiated cells declines as development proceeds, it is not clear whether glycolytic activity is temporally regulated within RPCs and/or in other retinal cell types. To assess glycolytic gene expression within specific cell populations (*Figure 1A*), we mined scRNA-seq data collected from developing retinas between E11 and P14 (*Clark et al., 2019*). High-dimensional uniform manifold approximation and projection plots identified unique cell clusters corresponding to pre-neurogenic (pre-N) RPCs at E11, early-RPCs from E12 to E18, and late-RPCs from P0 to P5 (*Figure 1B, C*). Feature plots were used to examine glycolytic gene expression in individual retinal cell types (*Figure 1C*). Several glycolytic genes were specifically enriched in early compared to late-RPC clusters, including *Pgk1*, a rate-limiting enzyme that catalyzes the first ATP-producing step in the glycolytic chain reaction, and *Eno1*, another rate-limiting enzyme (*Figure 1C*). A less obvious bias toward early-RPC expression was observed for *Hk1*, the initial enzyme in the glycolytic pathway that converts glucose to glucose-6-phosphate, *Pfk1*, another rate-limiting enzyme, and *Slc16a3*, encoding the lactate/H$^+$ symporter that shuttles lactate together with a H$^+$ ion (*Figure 1C*). In contrast, *Aldoa*, *Gpi1*, and *Ldhb* showed similar expression levels between pre-N-, early-, and late-RPCs (*Figure 1C*). These data confirm that glycolytic gene expression is elevated in RPCs and may peak during the early-to-late-RPC transition.

We noted that the highest levels of *Pgk1* expression at E18/P0 corresponded to the known pinnacle of RPC differentiation into committed, rod photoreceptor precursors (*Kim et al., 2016*; *Figure 1C*), which undergo a protracted differentiation period, only expressing mature photoreceptor markers between P2 and P8 (e.g., *Prph2*, *Impg2*, *Pde6g*, *Gm11744*, and *Gnat2*; *Figure 1—figure supplement 1*). We thus asked whether levels of glycolytic activity have an instructive role in rod photoreceptor fate selection. To test this hypothesis, we used two genetic models to elevate glycolytic activity in RPCs. We first targeted PTEN, a phosphatase that dephosphorylates and inactivates PGK1, a rate-limiting step (*Zhang et al., 2023b*). *Pten* is expressed throughout retinal development (*Figure 1—figure*

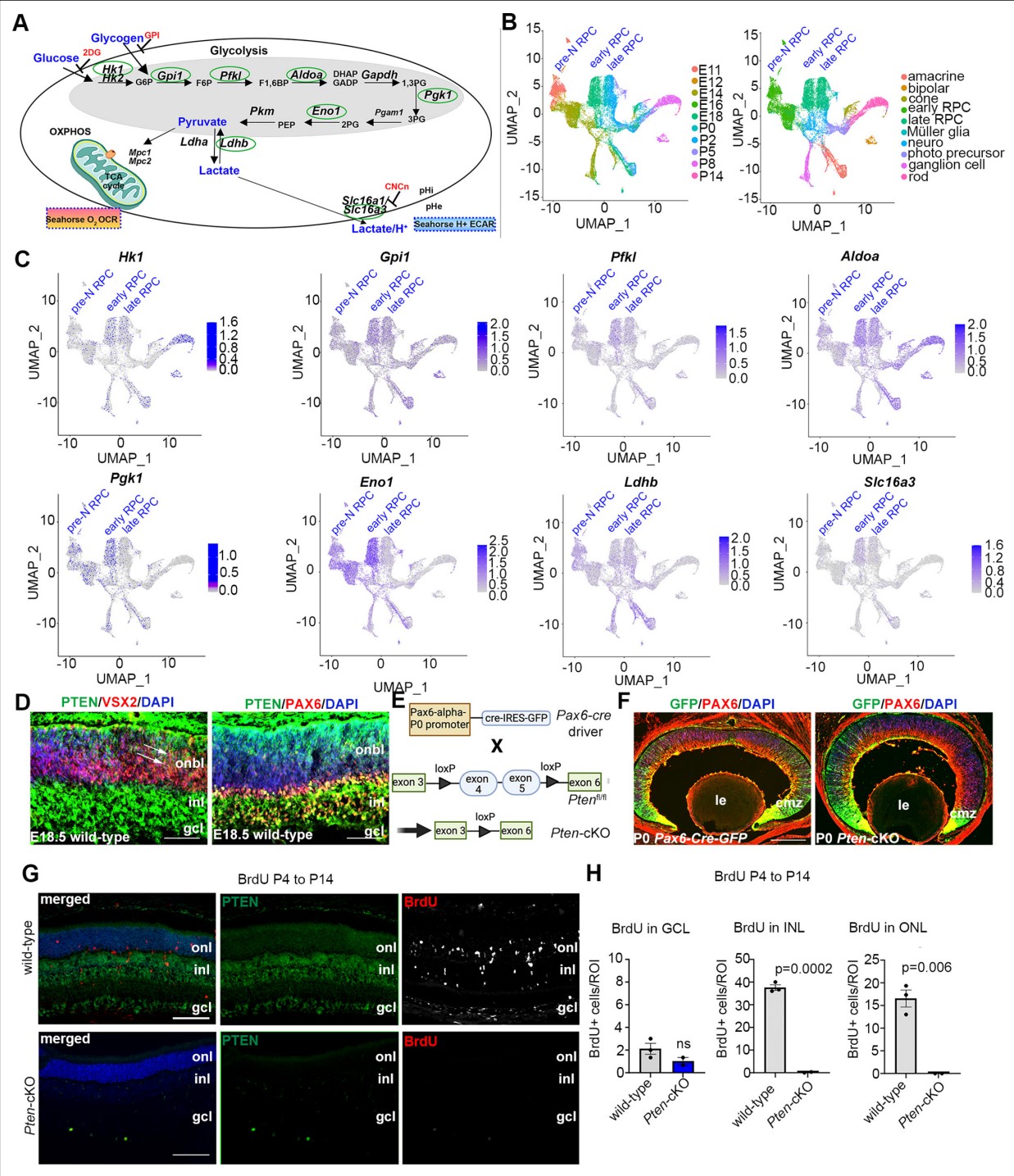

**Figure 1.** Glycolysis expression during development and accelerated development in *Pten*-cKO retinas. (**A**) Schematic of the key enzymatic steps in glycolysis. (**B**) Uniform manifold approximation and projection (UMAP) plot of scRNA-seq data collected from wild-type retinas between E11 and P14 (**Clark et al., 2019**). Stages of data collection are color-coded in the first panel, and cell type annotation is color coded in the second panel. Pre-neurogenic (N)-retinal progenitor cell (RPC), early-RPC, and late-RPC clusters are labeled. (**C**) Feature plots showing the transcript distribution of glycolytic genes, including *Pgk1*, *Eno1*, *Gpi1*, *Hk1*, *Pfkl*, *Slc16a3*, *Aldoa*, and *Ldhb*, showing enriched *Pgk1*, *Eno1*, and *Hk1* expression in earlier-staged RPCs. Pre-neurogenic (N)-, early-, and late-RPC clusters are labeled. (**D**) E18.5 retinal cross sections immunostained with PAX6, PTEN, and VSX2. (**E**) Schematic of the strategy used to generate *Pten*-cKO animals. (**F**) Immunostaining of P0 Pax6-Cre-GFP and *Pten*-cKO retinas, showing GFP expression in RPCs throughout the retina, with higher levels in the periphery in the ciliary marginal zone (CMZ). Created in BioRender.com. (**G**) Retinal cross sections of wild-type and *Pten*-cKO animals at P14 injected with BrdU at P4, showing immunostaining with PTEN (green) and BrdU (red) in wild-type and *Pten*-cKO retinas. (**H**) Quantification of the number of BrdU cells in the gcl, inl, and onl. Plots show means ± SEM. *N* = 3 biological replicates/genotype, all

*Figure 1 continued on next page*

*Figure 1 continued*

with 3 technical replicates. p-value calculated with unpaired *t*-test. onl, outer nuclear layer; inl, inner nuclear layer; gcl, ganglion cell layer; onbl, outer neuroblast layer; inl, inner nuclear layer; gcl, ganglion cell layer; le, lens; cmz, ciliary marginal zone. Scale bar 50 µM in C, 400 µM in E and 100 µM in F.

The online version of this article includes the following figure supplement(s) for figure 1:

**Figure supplement 1.** Expression of *Pten* and mature photoreceptor markers during murine retinal development.

*supplement 1*), including in late-RPCs (E18.5), which co-express VSX2 and PAX6 (*Figure 1D*). To delete *Pten* in RPCs, we generated a *Pten*-cKO model by crossing a floxed allele of *Pten*, in which exons 4 and 5 were flanked by loxP sites, with a *Pax6-Cre-GFP* driver allele (*Figure 1E*). As a proxy of Cre activity, GFP was expressed in RPCs and postmitotic newborn cells with a high peripheral to central gradient in *Pax6-Cre-GFP* transgenics (*Figure 1F*). Thus, for the remainder of this study, we focused on the peripheral retina where the *Pax6-Cre* allele was most active, avoiding the central retina, which was readily distinguishable as it forms a visible hamartoma comprised of remaining wild-type cells in *Pten*-cKO retinas (*Tachibana et al., 2018*).

Using cellular birthdating, we previously showed that *Pten*-cKO RPCs are hyperproliferative and differentiate on an accelerated schedule between E12.5 and E18.5, yet fewer rod photoreceptors are ultimately present in P7 (*Tachibana et al., 2016*) and P21 (*Hanna et al., 2025*) retinas, suggestive of a developmental defect. Since retinal cell differentiation persists until ~P12, we conducted similar birthdating experiments, but instead administered BrdU to P4 pups. In wild-type retinas harvested at P14, BrdU cells were mainly distributed between the INL, where late-born bipolar cells and Müller glia localize, and the ONL, where rod photoreceptors migrate (*Figure 1G, H*). Labeled cells in the ganglion cell layer (GCL) likely represent displaced amacrine cells, which are at the end of their differentiation window. In contrast, only a few BrdU-labeled cells were detected in the GCL in *Pten*-cKO retinas, with no BrdU$^+$ cells present in the INL and ONL (*Figure 1G, H*). Thus, the accelerated differentiation observed during embryonic stages is followed by the precocious depletion of dividing RPCs in *Pten*-cKO retinas by the early postnatal period, effectively limiting the overall period of rod photoreceptor differentiation (*Figure 1G, H*).

## Glycolytic gene expression and activity are upregulated in P0 *Pten*-cKO retinas

To investigate how *Pten*, encoding an intracellular phosphatase, controls the timing of RPC proliferation and differentiation, we began with a transcriptomic approach. To decipher transcriptomic changes associated with *Pten* loss, we performed bulk RNA sequencing (RNA-seq) on P0 wild-type (*N* = 4) and *Pten*-cKO (*N* = 5) retinas. A heatmap depiction revealed that there are distinct blocks of expressed genes in the two genotypes (*Figure 2A*; *Figure 2—source data 1*). Among the differentially expressed genes (DEGs) with a log$_2$-fold change (log2FC) ≥2 between wild-type and *Pten*-cKO retinas, there were 408 downregulated and 667 upregulated genes (*Figure 2B*). To draw relationships between genes and pathways, we analyzed the Kyoto Encyclopedia of Genes and Genomes (KEGG) database, showing enrichment of terms such as *axon guidance* and *regulation of actin cytoskeleton* among the DEGs. In addition, metabolic pathway terms were enriched, including *glycolysis* (*Figure 2C*), which involves a series of enzymatic reactions that break down glucose into pyruvate for energy production (*Figure 2D*). Indeed, analysis of Reads Per Kilobase per Million mapped reads (RPKM) revealed that most of the glycolytic regulatory enzymes were upregulated in *Pten*-cKO retinas, including *Hk1*, *Gpi1*, *Pfkl*, *Aldoa*, *Pgk1*, *Pgam1*, and *Eno1*, although others, such as *Pkm*, were not differentially expressed (*Figure 2E*). Notably, since we employed 75 bp single-end sequencing, we could not distinguish between alternatively spliced *Pkm1* and *Pkm2* mRNAs, bearing a common 3′ end. Additionally, expression of *Slc16a3*, encoding a monocarboxylate transporter that symports lactate and one hydrogen ion outside of the cell, was increased in *Pten*-cKO retinas (*Figure 2E*). We validated these increases in gene expression for a handful of genes, including *Hk1*, *Pgk1*, *Eno1*, and *Slc16a3*, using qPCR, confirming that these glycolytic genes were indeed upregulated in E15.5 *Pten*-cKO retinas (*Figure 2F*).

To test whether transcriptomic changes in P0 *Pten*-cKO retinas reflected an increase in glycolytic activity, we performed a Seahorse assay. Extracellular acidification rate was elevated in P0 *Pten*-cKO RPCs compared to controls, indicative of an increase in lactic acid levels, which serves as a proxy

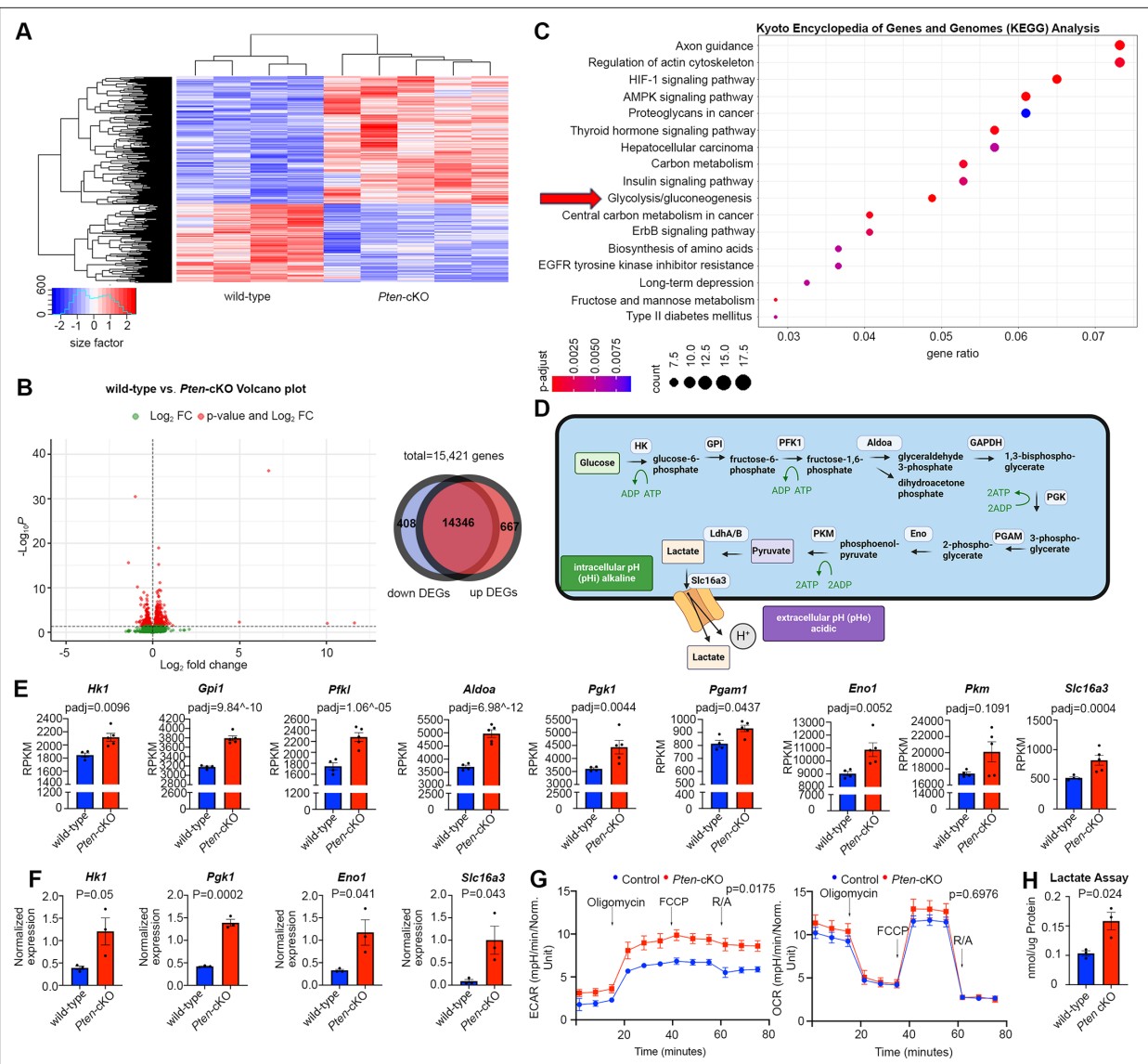

**Figure 2.** Glycolytic gene expression and activity are elevated in P0 *Pten*-cKO retinas. (**A**) Heatmap of bulk RNA-seq data comparing gene expression in P0 *Pten*-cKO ($N = 5$) and wild-type ($N = 4$) retinas. (**B**) Volcano plot of differentially expressed genes (DEGs) with log$_2$FC = 2, showing both downregulated (to the left) and upregulated (to the right) genes. A Venn diagram shows 667 upregulated genes and 408 downregulated genes in P0 *Pten*-cKO retinas. (**C**) Kyoto Encyclopedia of Genes and Genomes (KEGG) pathway enrichment related to DEGs, showing an enrichment of upregulated glycolytic pathway genes in P0 *Pten*-cKO retinas (arrow). (**D**) Schematic of glycolysis pathway, showing the intracellular enzymes involved in metabolizing glucose into lactate. Also shown is the Slc16a3 symporter, which extrudes lactate and H$^+$ to increase intracellular pHi. Created in BioRender.com. (**E**) Normalized RPKM (reads per kilobase million) values for glycolytic gene expression in P0 wild-type and *Pten*-cKO retinas. Plots show means ± SEM. $N = 4$ biological replicates for wild-type and $N = 5$ for *Pten*-cKO retinas. p-values calculated with Wald test, with a Benjamini–Hochberg correction for multiple comparisons. (**F**) Normalized glycolytic gene expression from qPCR of P0 *Pten*-cKO ($N = 3$) and wild-type control ($N = 3$) retinas. Plots show means ± SEM. $N = 3$ biological replicates/genotype, all with 3 technical replicates. p-values calculated with unpaired *t*-test. (**G**) Seahorse assay on P0 wild-type and *Pten*-cKO retinal progenitor cells (RPCs) cultured in vitro. Points of treatment with oligomycin (O), FCCP (F), and Rotenone/Antimycin A (R/A) are indicated. Left plot shows elevated extracellular acidification rate (ECAR) in *Pten*-cKO RPCs, while plot to the right shows no significant effect on oxygen consumption rate (OCR). Plots show means ± SEM. $N = 3$ biological replicates for *Pten*-cKO and $N = 8$ biological replicates for wild-type, all with 3 technical replicates. p-values calculated with unpaired *t*-test (GraphPad Prism, USA). (**H**) Lactate assay on P0 wild-type and *Pten*-cKO retinas. Plots show means ± SEM. $N = 3$ biological replicates/genotype, all with 3 technical replicates. p-value calculated with unpaired *t*-test.

The online version of this article includes the following source data for figure 2:

**Source data 1.** Identification of differentially expressed genes in P0 *Pten*-cKO retinas.

measure for glycolysis (*Figure 2G*). We further assessed mitochondrial $O_2$-consumption rates in P0 retinal cells, showing that oligomycin, an ATP synthase inhibitor, suppressed basal respiration rates to the same extent in wild-type and *Pten*-cKO retinas (*Figure 2G*). Similarly, maximum or spare respiratory capacity measured upon stimulation did not differ between P0 wild-type and *Pten*-cKO retinas (*Figure 2G*). Finally, we used a colorimetric assay to measure lactate levels in tissue lysates, revealing an elevation of lactate in *Pten*-cKO retinas compared to wild-type retinas at P0 (*Figure 2H*). Taken together, these data indicate that glycolytic gene expression and activity are elevated in *Pten*-cKO retinas, whereas oxygen-dependent OXPHOS is unaffected.

## Pharmacological inhibition of glycolysis decreases RPC proliferation and photoreceptor differentiation

Glucose is transported across cell membranes via glucose transporters (GLUT), with *Slc2a1*, encoding GLUT1, expressed at the highest levels in dividing, $Cdk1^+$ RPCs (*Figure 3A*). To assess the requirement for glycolysis during retinal development, we developed an in vitro retinal explant assay, dissecting retinas from P0 eyes and removing all non-retinal tissues, including the retinal pigment epithelium (RPE) (*Figure 3B*). P0 explants were mounted on Nucleopore membranes and cultured on top of retinal explant media, providing a source of nutrients, growth factors, and glucose. To block glycolysis, retinal explants were cultured with 2-deoxy-D-glucose (2DG), a competitive inhibitor that is transported by GLUT1 and converted by HK1/2 to 2-deoxy-D-glucose-6-phosphate, an inhibitor of the glycolytic pathway (*Figure 3C*). 2DG is typically used in the range of 5–10 mM in cell culture studies and in general, has anti-proliferative effects (*Zhang et al., 2014*). To test whether 5 mM- and 10 mM-2DG impacted RPC proliferation, explants were exposed to BrdU, which is incorporated into S-phase cells, for 30 min prior to harvesting. 2DG treatment resulted in a dose-dependent inhibition of RPC proliferation as evidenced by a reduction in $BrdU^+$ cells (*Figure 3D*). To determine whether glycolytic inhibition reduced the overall percentage of RPCs that continued to proliferate, we measured the proliferative fraction (p-fraction) by exposing retinal explants to BrdU at the onset of 2DG treatment, and then immunostaining for BrdU and Ki67, which labels all cycling cells. The p-fraction was calculated as the ratio of $Ki67^+$ cells that were $BrdU^+$ (i.e., %$BrdU^+Ki67^+$/$Ki67^+$ cells). The total numbers of $BrdU^+$ and $Ki67^+$ cells declined in 2DG-treated explants, as did the overall p-fraction (*Figure 3E*), indicating that our 2DG treatment was in the effective range.

A reduction in the p-fraction could indicate that RPCs had stopped cycling during the 24 hr window of 2DG treatment, or that these cells had undergone apoptosis. To distinguish between these possibilities, we first immunostained 2DG-treated retinal explants with VSX2, which primarily labels RPCs at P0. 2DG treatment for 24 hr did not alter the total number of $VSX2^+$ RPCs, even at the higher dose of 10 mM, despite the dramatic decline in proliferating RPCs (*Figure 3F*). Furthermore, 2DG treatment did not elevate levels of apoptosis, as the number of retinal cells expressing cleaved-caspase 3 (CC3), an executioner protease, remained similar in 2DG-treated and control explants (*Figure 3G*). We then asked whether glycolytic inhibition impacted cellular differentiation, focusing on CRX-expressing photoreceptor precursors, which are the main cells generated at P0. At the highest 10 mM 2DG dose, the number of $CRX^+$ cells declined, indicative of a dose-dependent decrease in the number of photoreceptor precursors born (*Figure 3H*). Taken together, these data indicate that glycolytic inhibition suppresses RPC proliferation, and at higher doses, prevents RPCs from differentiating into photoreceptors. Stated differently, glycolysis is required to support RPC proliferation and photoreceptor differentiation.

Previous studies in *Xenopus* similarly found that RPCs rely mainly on glycolysis for energy production, however, instead of the demand for a constant supply of glucose, *Xenopus* RPCs depend on glycogen storage (*Agathocleous et al., 2012*). To test whether mouse RPCs similarly require stored glycogen, P0 retinal explants were cultured in the presence of 12.5 and 25 μM glycogen phosphorylase inhibitor (GPI) for 24 hr (*Figure 3I*), the latter corresponding to the same concentrations used in *Xenopus* retina (*Agathocleous et al., 2012*). In murine retinal explants, GPI treatment did not impact the number of RPCs that incorporated BrdU (*Figure 3I*). These data suggest that rather than depending on glycogen stores, murine RPCs depend on a constant supply of glucose to maintain proliferation, as shown in human retinal organoids (*Takata et al., 2023*).

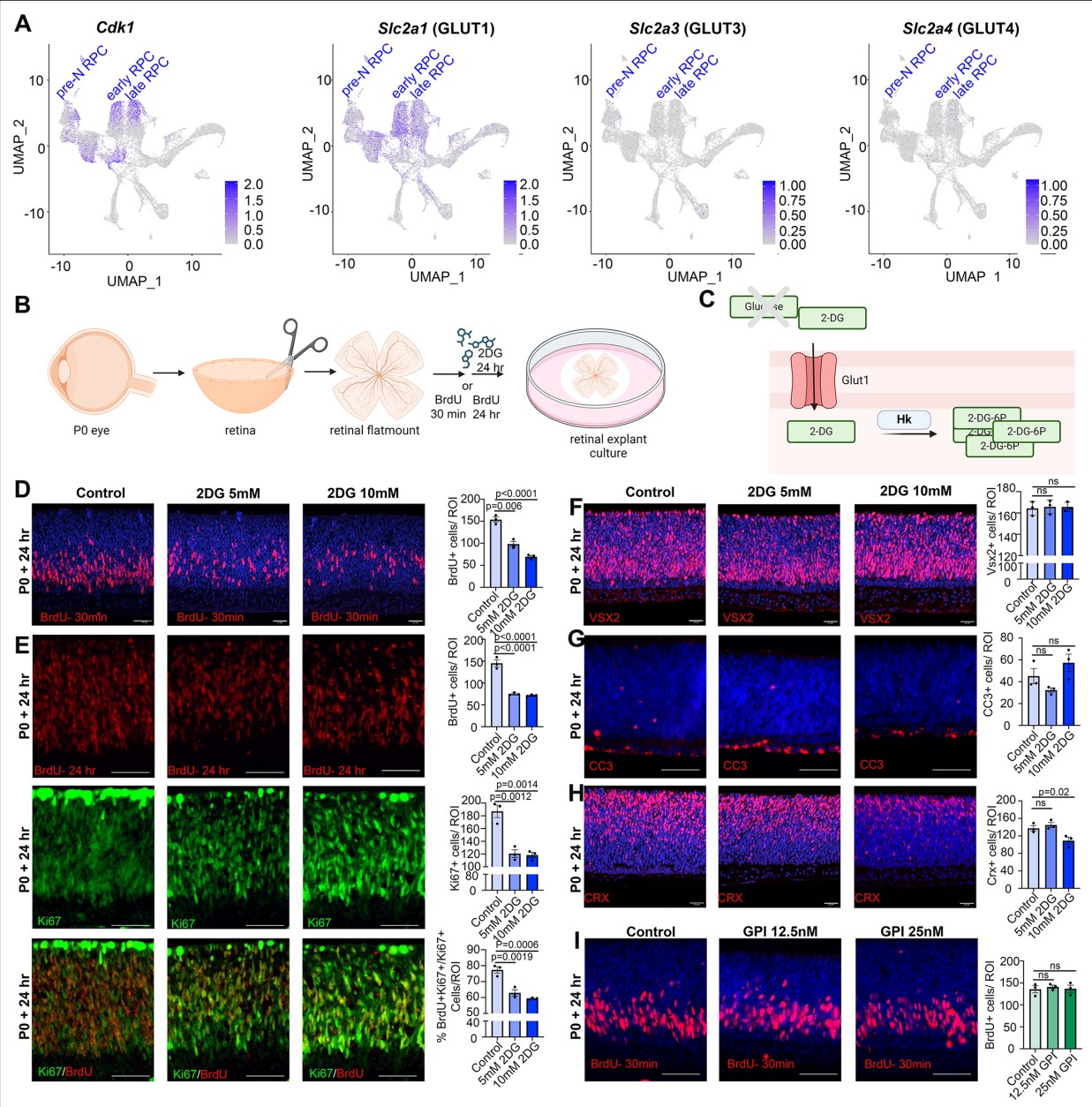

**Figure 3.** Glycolytic inhibition reduces proliferation and photoreceptor differentiation in P0 retinal explants. (**A**) Feature plots showing the transcript distribution of *Cdk1*, a marker of proliferating retinal progenitor cells (RPCs), and the genes encoding the major glucose transporters, *Slc2a1* (GLUT1), *Slc2a3* (GLUT3), and *Slc2a4* (GLUT4). (**B**) Schematic of retinal explant experiments, showing treatment with pharmacological inhibitors (2DG or GPI), and the labeling of proliferating RPCs (30 min BrdU) or birthdating of newborn retinal cells (24 hr BrdU). Created in BioRender.com. (**C**) Schematic of glucose uptake replaced by 2DG, which leads to the accumulation of 2-DG-6P that cannot be further metabolized by the glycolytic pathway. Created in BioRender.com. (**D**) P0 retinal explants treated with 5 or 10 mM 2DG for 1 day in vitro (DIV), showing BrdU immunolabeling and the number of BrdU$^+$ cells after a 30 min labeling with BrdU. (**E**) P0 retinal explants treated with 5 or 10 mM 2DG for 1 DIV, showing BrdU immunolabeling and the number of BrdU$^+$ cells after a 24 hr labeling with BrdU, including co-labeling with Ki67 to calculate the p-fraction (%BrdU$^+$Ki67$^+$ cells/total Ki67$^+$ cells). (**F–H**) P0 retinal explants treated with 5 or 10 mM 2DG for 1 DIV and immunostained for VSX2 (**E**), CC3 (**F**), or CRX (**G**). (**I**) P0 retinal explants treated with glycogen phosphorylase inhibitor (GPI) at 12.5 and 25 μM for 24 hr and immunostained for BrdU after a 30-min incubation at the end of the treatment period. Plots in D–I show means ± SEM. *N* = 3 biological replicates/treatment, all with 3 technical replicates. p-values calculated with one-way ANOVA with Tukey post hoc test. Scale bars = 25 μM in D, F, H and 50 μM in E, G, I.

## Precocious rod differentiation and maturation in cytoPFKFB3 GOF mice phenocopies *Pten*-cKO retinal features

We previously showed that at P7, *Pten*-cKO retinas are comprised of fewer differentiated rods and amacrine cells, while other cell types are not affected (*Tachibana et al., 2016*). Given the increase in glycolysis in *Pten*-cKO retinas observed in the current study, we asked whether metabolic reprogramming may be sufficient to bias RPC fate choice, thereby altering cellular composition. As a precedent, glycolytic flux alters neuromesodermal lineage selection in the chick tail bud (*Oginuma et al., 2020*) and in mouse mesoderm (*Miyazawa et al., 2022*). To determine whether increased glycolysis is a critical downstream effector pathway of *Pten* in retinal cell fate selection, we used a transgenic approach to elevate glycolytic flux in RPCs. Glycolytic flux is defined as the conversion of fructose-6-phosphate (F6P) to fructose-1,6-bisphosphate (FBP), which is catalyzed by phosphofructokinase (PFK1), and is the first rate limiting, commitment step in glycolysis (*Miyazawa et al., 2022*; *Figure 4A*). To promote glycolytic flux, we used a conditional GOF transgenic line driving the expression of a dominant active, cytoplasmic form of PFKFB3 (cytoPFKFB3) (*Miyazawa et al., 2022*). PFKFB3 converts F6P to fructose-2,6-bisphosphate, which activates PFK1 (*Shi et al., 2017*; *Figure 4A*). CytoPFKFB3 mice harbor a floxed stop cassette in front of a mutant form of human PFKFB3 (K472A/K473A) that localizes to the cytoplasm and activates glycolysis (*Miyazawa et al., 2022*; *Yalcin et al., 2009*). To overexpress cytoPFKFB3 in RPCs, we removed the floxed stop cassette using a *Pax6-Cre* driver to create a cytoPFKFB3 GOF model (*Figure 4B*). Of note, the same *Pax6-Cre* driver was employed to generate *Pten*-cKOs used in this study. To confirm that cytoPFKFB3-GOF retinas undergo increased glycolytic flux, we measured levels of secreted lactate as a proxy measure using a colorimetric assay. We observed an overall increase in lactate levels in P0 cytoPFKFB3-GOF retinas (*Figure 4C*).

Having validated the GOF model, we next asked whether elevating cytoPFKFB3 expression in RPCs impacted the timing of retinal cell differentiation using a birthdating assay. BrdU was administered to pregnant females at E12.5 and pups were sacrificed at P7. In wild-type retinas, BrdU$^+$ cells were mainly distributed in the GCL, where newborn ganglion cells and displaced amacrine cells reside, and in the INL, where amacrine cells and horizontal cells are located (*Figure 4D, E*). Fewer BrdU$^+$ cells labeled at E12.5 were present in the P7-ONL, with these early-born ONL cells primarily including cones in wild-type retinas (*Figure 4D, E*). In cytoPFKFB3-GOF retinas, BrdU$^+$ cells were also distributed among the three layers, but BrdU$^+$ cells were more abundant in the ONL, where photoreceptors reside, with a concomitant reduction in the INL (*Figure 4D, E*). The shift in distribution of E12.5-labeled, BrdU$^+$ cells phenocopies the altered temporal differentiation observed in *Pten*-cKO retinas (*Tachibana et al., 2016*).

The accelerated differentiation of ONL cells in cytoPFKFB3-GOF retinas prompted us to ask whether photoreceptor maturation was similarly hastened, focusing on OS formation as a readout. While most differentiated cells use OXPHOS for energy production, photoreceptors are an important exception, as these metabolically demanding cells require glycolysis to maintain OSs, which undergo diurnal turnover (*Chinchore et al., 2017*). To examine OS formation, we immunostained P7 wild-type and cytoPFKFB3-GOF retinas with rhodopsin (Rho) (*Figure 4F*). Visually, photoreceptor OS above the ONL, which are labeled with Rho, were larger in P7 cytoPFKFB3-GOF retinas, as confirmed by a measurement of the OS area (*Figure 4F*).

To determine whether rod photoreceptor OSs similarly mature precociously in *Pten*-cKO retinas, we first examined the P0 transcriptomic data for OS-enriched genes (*Datta et al., 2015*). We identified several OS genes that were upregulated in P0 *Pten*-cKO retinas, although not all, as several OS genes were also downregulated (*Figure 4G*). To assess OS maturation morphologically, we immunostained P7 retinas with Rho, revealing a similar increase in the rhodopsin-labeled OS area in *Pten*-cKOs (*Figure 4I*). Finally, to investigate the role of glucose metabolism in accelerating the rate of rod photoreceptor maturation, we inhibited glycolysis during the early postnatal window by in vivo administration of 2DG, which was administered daily from P0 to P7 (*Figure 4H, I*). Strikingly, 2DG treatment blocked the precocious maturation of rod photoreceptor OSs in *Pten*-cKO retinas, which returned to wild-type sizes (*Figure 4I*). Taken together, these data implicate glucose metabolism and glycolysis as an important driver of photoreceptor differentiation and OS maturation.

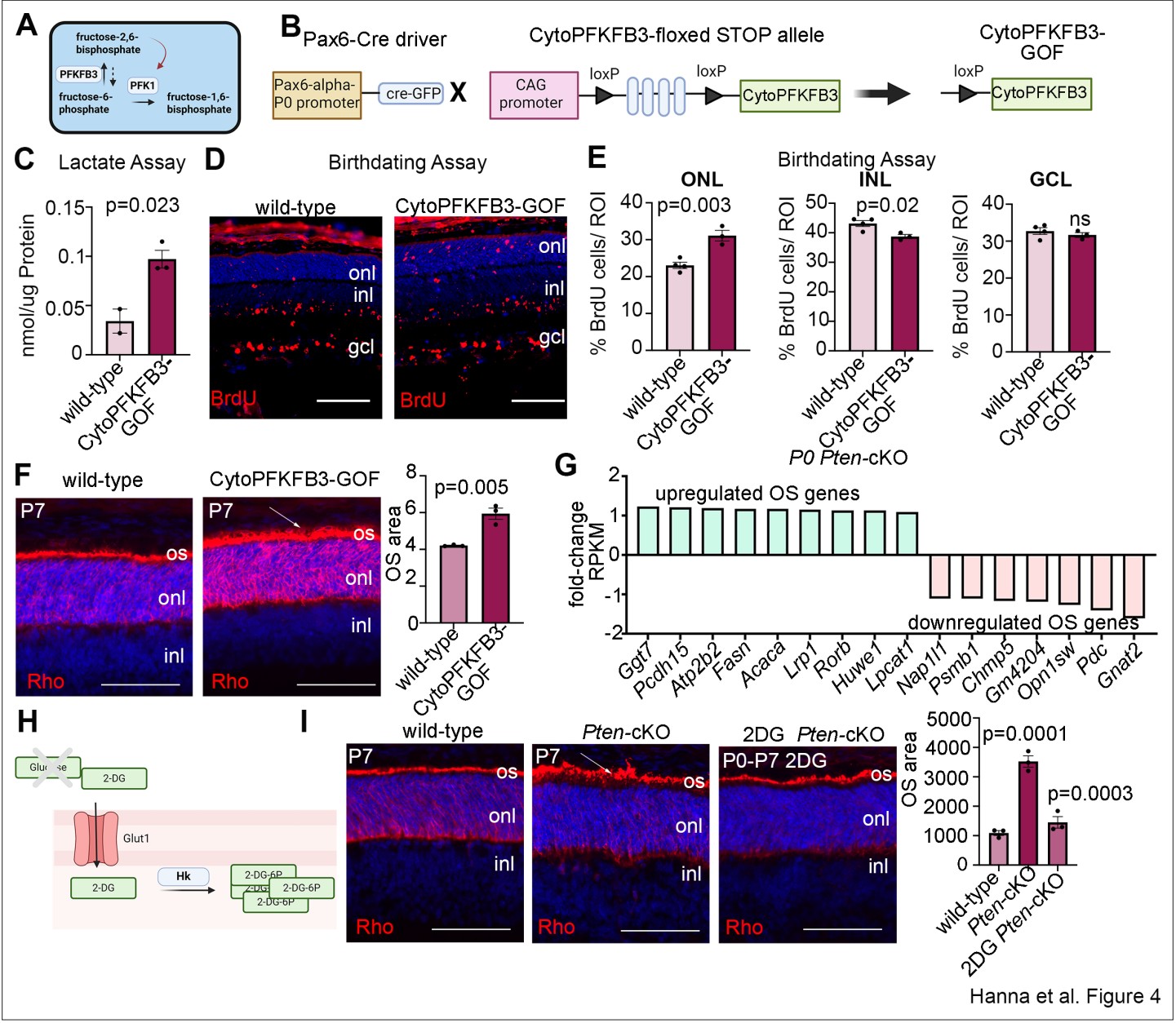

**Figure 4.** Promotion of glycolytic flux with a cytoPFKFB3-GOF mouse model accelerates photoreceptor differentiation and outer segment (OS) maturation. (**A**) Schematic showing function of PFKFB3 as an activator of PFK1 through the conversion of fructose-6-phosphate to fructose-2,6-bisphosphate. Created in BioRender.com. (**B**) Schematic showing generation of a retinal progenitor cell (RPC)-specific cytoPFKFB3-GOF mouse model. Created in BioRender.com. (**C**) Lactate assay performed on P0 wild-type and cytoPFKFB3-GOF retinas. (**D, E**) Birthdating experiments performed by injecting BrdU into pregnant females at E12.5 and harvesting retinas at P7. BrdU immunostaining of P7 wild-type and cytoPFKFB3-GOF retinas (**D**) and quantification of the %BrdU+ cells in each of the nuclear layers (**E**). (**F**) Rhodopsin immunostaining of P7 wild-type and cytoPFKFB3-GOF retinas. An expansion of OS area is indicated by the arrow and quantified in the graph. (**G**) Bulk RNA-seq mining of photoreceptor OS genes de-regulated in P0 *Pten*-cKO retinas. (**H**) Schematic of mode of action of 2DG. Created in BioRender.com. (**I**) Rhodopsin immunostaining of P7 wild-type and *Pten*-cKO retinas, either without treatment or following the administration of 2DG between P0 and P7. The expanded OS area in *Pten*-cKO retinas is indicated with an arrow. Quantifications of photoreceptor OS areas are shown in the graph. Plots show means ± SEM. $N$ = 3 biological replicates/treatment group except C, with cytoPFKFB3-GOF ($N$ = 3) and wild-type ($N$ = 2), all with 3 technical replicates. p-values calculated with *t*-tests in C and F, and one-way ANOVA with Tukey post hoc test in E and G. Scale bars 100 µM in E and 50 µM in F and G. gcl, ganglion cell layer; inl, inner nuclear layer; onl, outer nuclear layer; os, OSs.

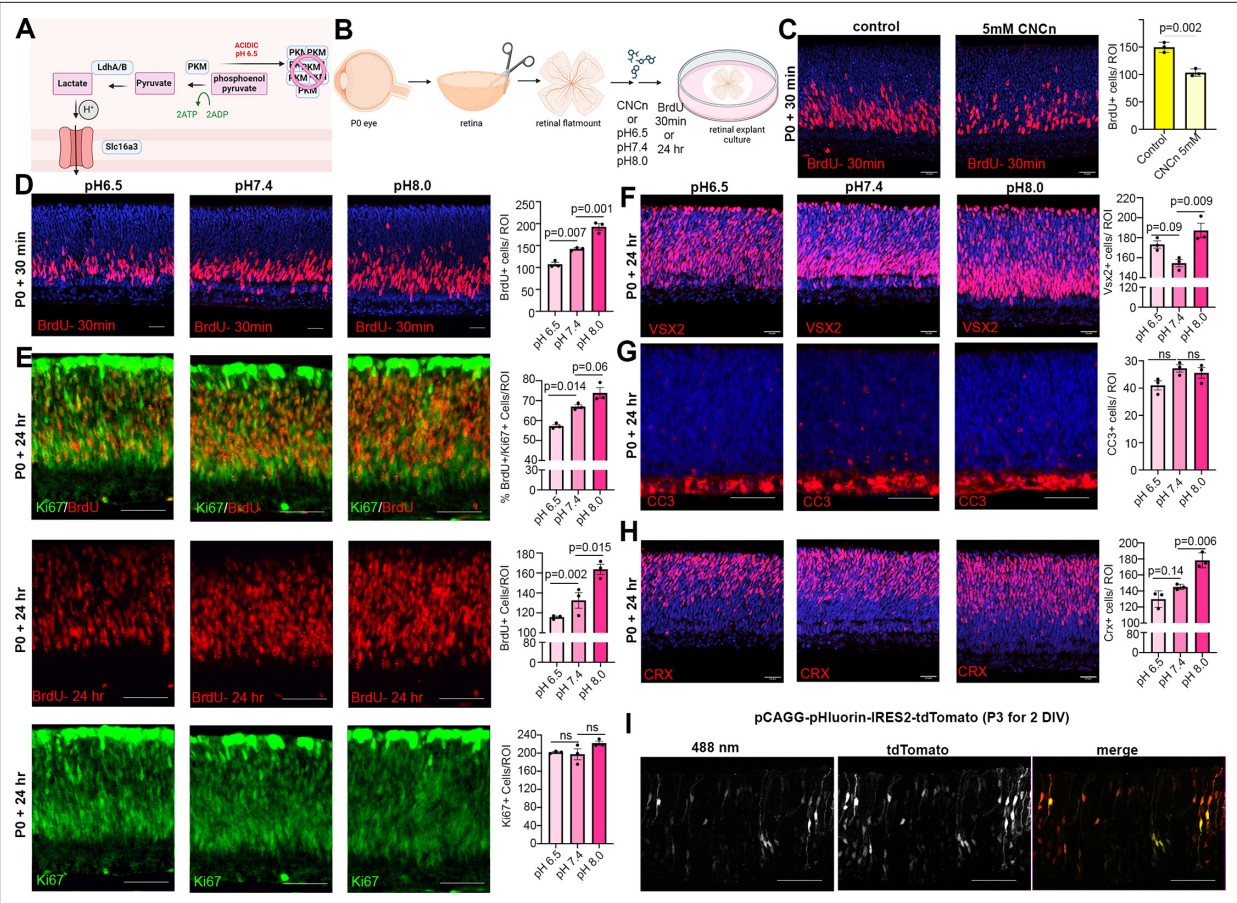

**Figure 5.** Reducing intracellular pH (pHi) inhibits retinal progenitor cell (RPC) proliferation and differentiation. (**A**) Schematic showing function of Slc16a1/3 symporters in extruding lactate and H⁺ out of the cell to increase pHi during elevated glycolysis. Also depicted is the impact of acidic pH and oligomerizing PKM to prevent its activity. Created in BioRender.com. (**B**) Schematic of retinal explant experiments, showing treatment with a pharmacological inhibitor (CNCn) of Slc16a symporters or growth in different pH media, and the labeling of proliferating RPCs (30 min BrdU) or birthdating of newborn retinal cells (24 hr BrdU). Created in BioRender.com. (**C**) P0 retinal explants treated with 5 mM CNCn for 1 day in vitro (DIV), followed by a 30 min BrdU pulse, showing immunolabeling and the quantification of BrdU⁺ RPCs. (**D**) P0 retinal explants incubated in media buffered to pH 6.5, pH 7.4, or pH 8.0 for 1 DIV, showing BrdU immunolabeling and the number of BrdU⁺ cells after a 30 min labeling with BrdU. (**E**) P0 retinal explants incubated in media buffered to pH 6.5, pH 7.4, or pH8.0 for 1 DIV, showing BrdU immunolabeling and the number of BrdU⁺ cells after a 24 hr labeling with BrdU, including co-labeling with Ki67 to calculate the p-fraction (%BrdU⁺Ki67⁺ cells/total Ki67⁺ cells). (**F–H**) P0 retinal explants incubated in media buffered to pH 6.5, pH 7.4, or pH 8.0 for 1 DIV and immunostained for VSX2 (**E**), CC3 (**F**), or CRX (**G**). (**I**) P3 retinal explant electroporated with pHluorin-tdTomato, imaged after 2 DIV to show electroporated cells (tdTomato⁺) have varying levels of 488 nm emission from pHLuorin (green). Plots show means ± SEM. *N* = 3 biological replicates/treatment group, all with 3 technical replicates. p-values calculated with *t*-tests in C, and one-way ANOVA with Tukey post hoc test in D–H. Scale bars = 25 µM in C, D, F, H, and 50 µM in E, G, I.

## Reducing pHi phenocopies the inhibitory effects of a glycolytic block on RPC proliferation and differentiation

Glycolysis is not an energy-efficient metabolic pathway as cells produce 2 ATP/glucose molecule versus 30–38 ATP/glucose when using OXPHOS (*Touahri et al., 2024*). However, cells rely on glycolysis for reasons other than energy expenditure. For instance, lactate is a metabolic byproduct that is shuttled out of the cell along with one H⁺ ion by lactate/H⁺ symporters (Slc16a group proteins) when glycolytic activity is high, effectively increasing intracellular pH (pHi) (*Figure 5A*). Notably, glycolysis-driven increases in pHi are the central driver of cell fate choice selection in the chick neuromesodermal lineage (*Oginuma et al., 2020*). We found *Slc16a* family lactate/H⁺ symporters are elevated in *Pten*^cKO retinas (*Figure 2E*), suggesting that more protons may be extruded to increase pHi. To address whether lactate/H⁺ symporters control RPC proliferation, we incubated wild-type P0 explants with 5 mM α-cyano-4-hydroxycinnamate (CNCn), an inhibitor of monocarboxylate transporters, including the Slc16a-family symporters, for 24 hr (*Figure 5B*). Inhibition of proton-linked lactate transport can

lead to intracellular acidification. Thus, the impact of CNCn should block the export of lactate into the extracellular space, effectively lowering pHi and further decreasing glycolysis. To determine whether RPC proliferation was impacted by CNCn treatment, explants were incubated with BrdU 30 min before harvesting. We observed a decrease in the number of BrdU+ proliferating RPCs in CNCn-treated retinal explants compared to controls (*Figure 5C*). This decline in RPC proliferation mirrored the effect of inhibiting glycolysis (*Figure 3C*). Thus, Slc16a-family symporters play a role in regulating RPC proliferation in the retina.

In keeping with the idea that pH levels influence cell fate selection, a previous study found that neuromesodermal progenitors in chick tailbud explants cultured in low pH media undergo cell cycle exit and lineage selection is biased toward neural rather than mesodermal fates (*Oginuma et al., 2020*). Given that altered glycolysis influences intracellular pH, which in turn controls cell fate decisions, we set out to assess the impact of manipulating pH on cell fate selection in the retina. One of the expected impacts of lowering pH was the aggregation of PKM2, a rate-limiting enzyme for glycolysis, which aggregates in reversible, inactive amyloids (*Cereghetti et al., 2024*). We grew retinal explants in media buffered at pH 6.5 (acidic), pH 7.4 (neutral), or pH 8.0 (basic). Increased glycolysis, leading to more H+ extrusion from the cell and an increase in pHi was modeled by pH 8.0, while decreased glycolysis, leading to less H+ extrusion and a lower pHi, was modeled by pH 6.5. To assess the impact of pHi changes on RPC proliferation, P0 retinal explants were cultured in media buffered to different pH levels, with BrdU added for 30 min and at the end of the 24-hr culture period. More RPCs were BrdU+ in pH 8.0 media compared to neutral pH 7.4 media, consistent with the pro-proliferative effect of increased glycolysis (*Figure 5D*). Conversely, in media buffered to pH 6.5, the number of BrdU+ proliferating RPCs declined (*Figure 5D*).

To determine whether pH changes alter the overall percentage of RPCs that continue to proliferate, we measured the p-fraction by exposing retinal explants to BrdU at the onset of pH adjustments and then calculating the p-fraction as the ratio of Ki67+ cells that were BrdU+. We found a pH-dependent increase in the number of BrdU+ RPCs and in the p-fraction in retinal explants cultured in pH 8.0 compared to pH 7.4 (neutral pH control) media (*Figure 5E*). We also observed a corresponding increase in VSX2+ RPCs in pH 8.0 media (*Figure 5F*), and more RPCs differentiated into CRX+ photoreceptor precursors (*Figure 5H*). In contrast, in retinal explants cultured in pH 6.5 media, fewer RPCs entered S-phase as evidenced by the decline in BrdU+ RPCs and lower p-fraction (*Figure 5E*). However, there were more VSX2+ RPCs in retinal explants grown in pH 6.5 versus pH 7.4 media (*Figure 5F*), with no effect on cell death evident as assessed by cleaved-caspase 3+ cell counts (*Figure 5G*), likely because fewer of these RPCs differentiated into CRX+ photoreceptor precursors (*Figure 5H*). Thus, pH has a dramatic effect on the decision by RPCs to proliferate versus differentiate, with higher pH more conducive to rapid, self-renewing, asymmetric divisions (P/D) that generate an RPC and a neuron.

Tissues naturally have pHi and differentiation gradients – for instance, there is an anterior (low pHi, more differentiated cells) to posterior (high pHi, more neuromesodermal progenitor cells) gradient in neuromesodermal progenitors in the embryonic chick tailbud (*Oginuma et al., 2020*). Given the importance of pH to control RPC proliferation and differentiation dynamics, and since not all RPCs differentiate at any given time, we asked whether there was a dynamic representation of pH within RPCs within the developing retina. To test this assumption, we electroporated P3 retinal explants with pCAGG-pHluorin-IRES2-tdTomato, using tdTomato to label transfected cells and pHluorin, a ratio-metric pH sensor with a 488 nm activation spectrum in neutral pH. As expected, the 488 nm signal was variable across RPCs that had similar tdTomato levels (*Figure 5I*). Taken together, this data is consistent with the idea that pH varies within RPCs, correlating with the propensity of some RPCs to continue to proliferate while others differentiate.

## Canonical Wnt signaling is active in neurogenic RPCs and controlled by glycolysis

Our finding that glycolysis might be a central driver of RPC fate selection, in part through the control of pHi, prompted the question of how these biochemical processes might impact cell fate selection. To glean further insights, we revisited our bulk RNA-seq dataset from P0 *Pten*-cKO retinas and performed a PANTHER (protein analysis through evolutionary relationships) classification analysis of DEGs (*Figure 6—figure supplement 1*). As expected, glycolysis was among the upregulated pathways, but the most upregulated pathway in P0 *Pten*-cKO retinas was Wnt signaling (*Figure 6—figure*

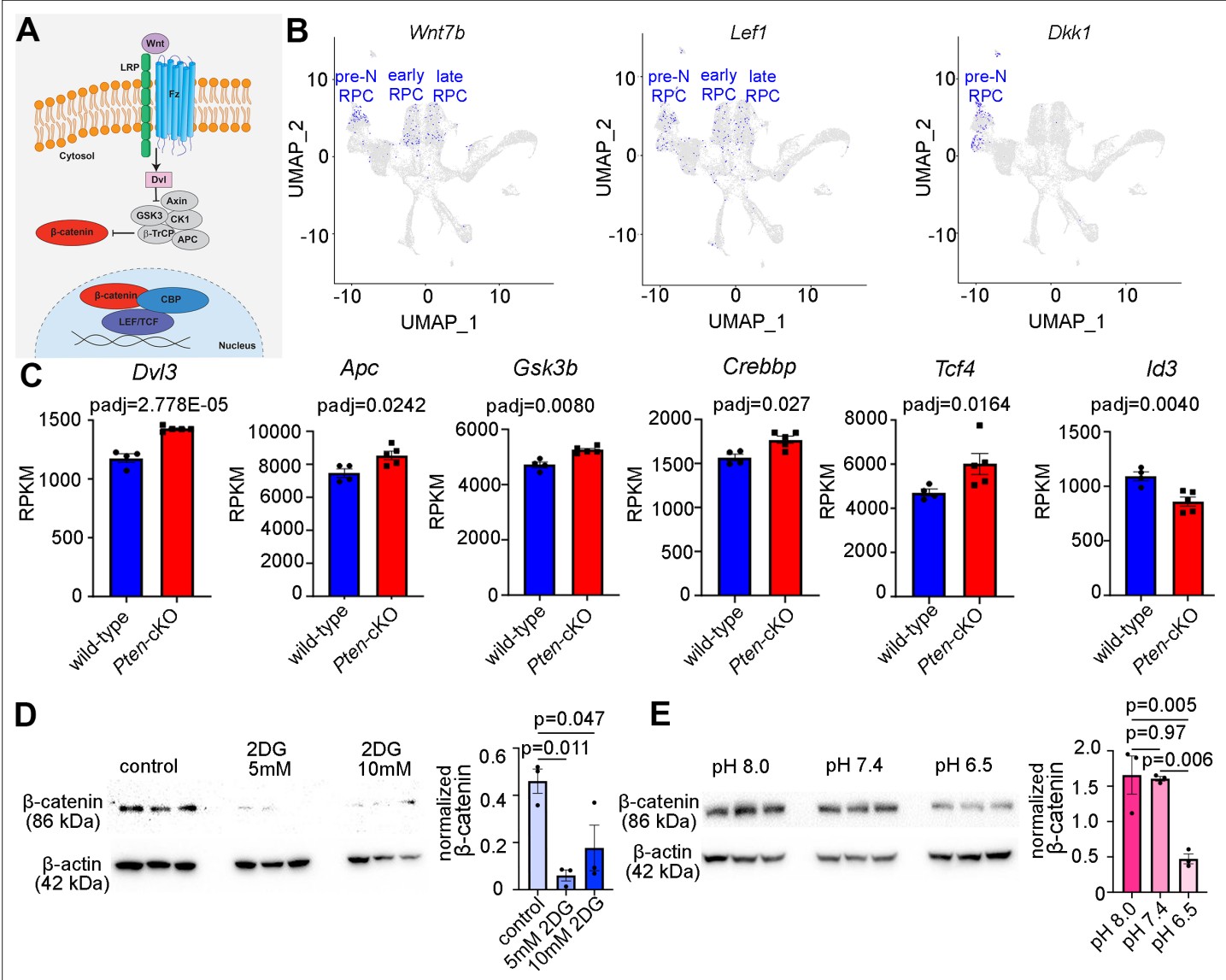

**Figure 6.** Wnt signaling is a downstream mediator of Pten and glycolysis. (**A**) Schematic illustration of canonical Wnt signaling pathway. (**B**) Uniform manifold approximation and projection (UMAP) plot of scRNA-seq data collected from wild-type retinas between E11 and P14 (*Clark et al., 2019*). Transcript distribution of *Wnt7b*, *Lef1*, and *Dkk1* Wnt genes. (**C**) Normalized RPKM (reads per kilobase million) values for Wnt pathway-associated genes in P0 wild-type (*N* = 4) and *Pten*-cKO (*N* = 5) retinas. (**D**) Western blot of β-catenin expression, normalized to β-actin levels, in P0 retinal explants treated with 5 and 10 mM 2DG or in P0 retinal explants cultured in media buffered to pH 8.0, pH 7.4, or pH 6.5 (**E**), all for 24 hr. Plots show means ± SEM. *N* = 3 biological replicates/treatment group, all with 3 technical replicates. p-values calculated with *t*-tests in C, and one-way ANOVA with Tukey post hoc test in D, E.

The online version of this article includes the following source data and figure supplement(s) for figure 6:

**Source data 1.** PDF composite file containing western blots for *Figure 6D, E*, indicating the treatments and the molecular weights of the relevant proteins.

**Source data 2.** Original tiff files for the western blots in *Figure 6D, E*.

**Figure supplement 1.** Upregulation of Wnt-associated genes in *Pten*-cKO retinas.

*supplement 1*). Wnts bind to frizzled (Fz) and low-density lipoprotein receptor-related protein receptors, which induces β-catenin translocation to the nucleus, where it forms an active transcriptional complex with T-cell-specific transcription factor (TF)/lymphoid enhancer-binding factor TFs (*Figure 6A*). Prior studies had implicated canonical Wnt signaling in maintaining an undifferentiated RPC state, especially in the peripheral margin at early developmental stages (*Ouchi et al., 2011*).

However, mining scRNA-seq data from E11 to P14 retinas (**Clark et al., 2019**) revealed that Wnt signaling may be active throughout retinal development, with transcripts for *Wnt7b* and *Lef1*, which is transcribed in response to canonical Wnt signaling (**Schmidt et al., 2000**), detected in RPCs at all developmental stages (**Figure 6B**). Furthermore, we noted that *Dkk*, a Wnt inhibitor, is only expressed in pre-neurogenic RPCs, and not in RPCs in the early and late neurogenic periods, suggesting that Wnt signaling may not be active prior to the onset of neurogenesis (**Figure 6B**). We thus explored whether Wnt signaling was regulated by glycolysis to play a role in RPC differentiation.

Components of the Wnt signaling pathway that were upregulated in P0 *Pten*-cKO retinas included *Dvl3*, an inhibitor of the β-catenin destruction complex, *Apc* and *Gsk3b*, two components of the destruction complex, and *Tcf4* and *Crebbp*, a downstream effector of Wnt signaling and its co-activator, respectively (**Figure 6C**). These data were consistent with our previous demonstration that canonical Wnt signaling levels are elevated in *Pten*-cKO retinas, as assessed by increased expression of non-phosphorylated (active)β-catenin (**Touahri et al., 2024**). However, unlike in the peripheral retina of *Ctnnb1*-cKOs (encoding for β-catenin), in which the neurogenic inhibitor *Id3* is activated by Wnt signaling in the periphery to maintain RPCs in an immature, undifferentiated state (**Ouchi et al., 2011**), we found that *Id3* is downregulated in P0 *Pten*-cKO retinas (**Figure 6C**), potentially allowing differentiation to proceed unimpeded.

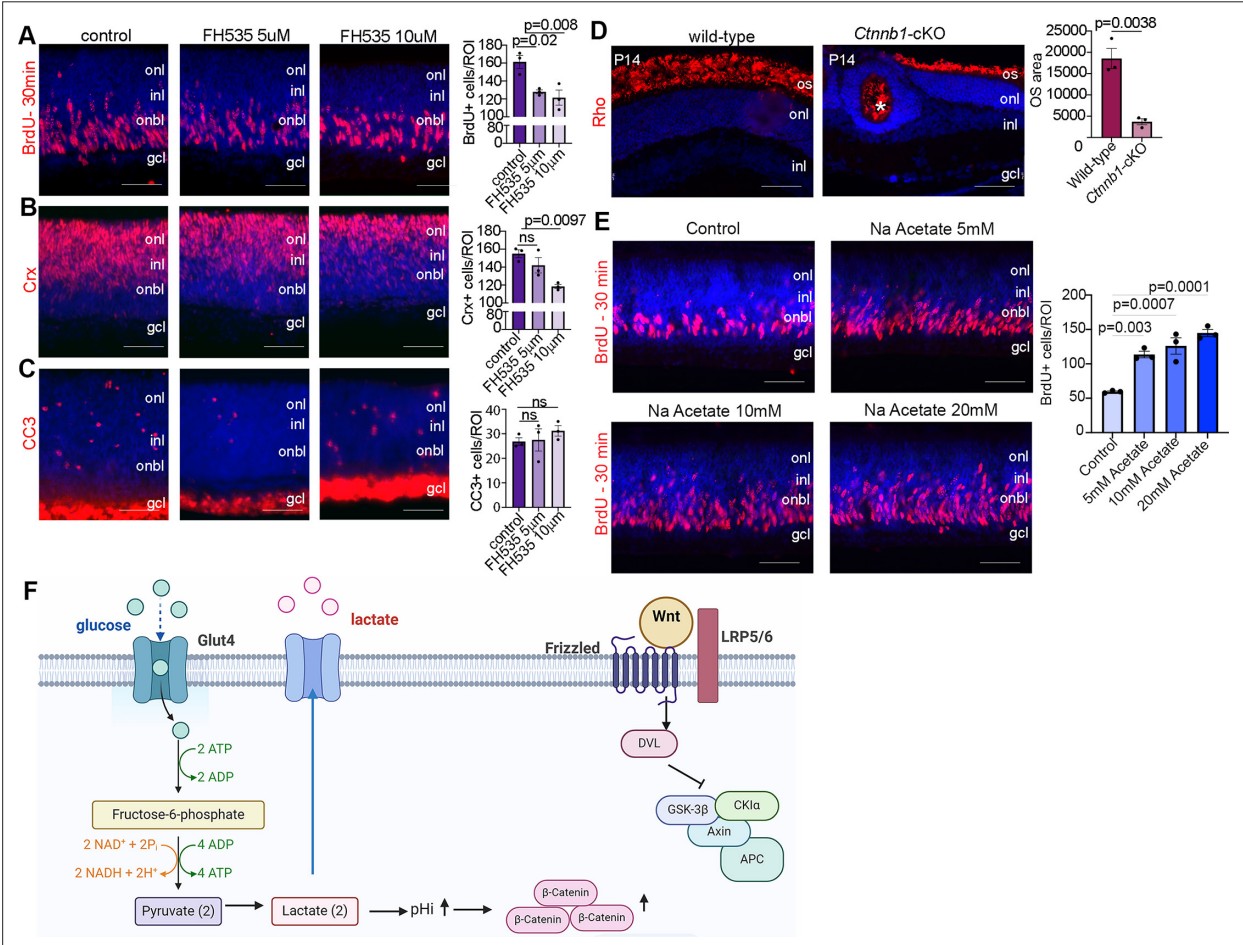

**Figure 7.** Wnt signaling modulates retinal proliferation and differentiation. (**A–C**) P0 retinal explants treated with 5 or 10 µM FH535 to block Wnt signaling for 1 day in vitro (DIV), followed by immunostaining for BrdU after a 30-min pulse (**A**), CRX (**B**), or CC3 (**C**). (**D**) Immunostaining P14 wild-type and *Ctnnb1*-cKO retinas with rhodopsin. Asterisk points to ONL rosette in *Ctnnb1*-cKO retina. Graph showing quantification of OS area. (**E**) Retinal cross sections of P0 wild-type explants treated with 5, 10, and 20 mM sodium acetate for 1 DIV, along with vehicle controls, and pulsed with BrdU 30 min before harvesting. The number of BrdU⁺ retinal progenitor cells (RPCs) increased in sodium acetate treated explants. (**F**) Summary schematic of the impact of glucose-driven glycolysis on intracellular pH and canonical Wnt signaling. Created in BioRender.com. All experiments included three biological replicates for each group. Statistical analysis was done using one-way ANOVA with Tukey post hoc test used in A–C and unpaired *t*-test in D. Scale bars = 50 µM in A–D. gcl, ganglion cell layer; inl, inner nuclear layer; onl, outer nuclear layer; os, outer segments.

Finally, to establish whether Wnt signaling was indeed regulated by glycolytic activity, we investigated β-catenin levels in explants treated with 2DG or grown in different pH media. β-Catenin levels declined upon glycolysis inhibition in 2DG-treated explants (*Figure 6D*), and at pH 6.5, a proxy for low glycolysis (*Figure 6E*). Thus, canonical Wnt signaling is elevated in P0 *Pten*-cKO retinas (*Touahri et al., 2024*), in which glycolysis is elevated, and reduced in conditions in which glycolysis is inhibited.

## Wnt signaling controls rod photoreceptor differentiation and maturation

We next asked whether inhibiting Wnt signaling would phenocopy the effects of reducing glycolysis on RPC proliferation and photoreceptor differentiation. For this purpose, we treated P0 wild-type CD1 retinal explants with 5 or 10 µM FH535, a pharmacological inhibitor of Wnt/β-catenin signaling (*Handeli and Simon, 2008*), for 24 hr. To label proliferating RPCs, BrdU was added 30 min before harvesting, similar to previous experiments. FH535-treated retinas showed a significant decrease in the number of BrdU$^+$ RPCs (*Figure 7A*) and a concomitant decrease in the number of CRX$^+$ photoreceptor precursors (*Figure 7B*) without impacting apoptosis (*Figure 7C*). To further study the role of Wnt signaling in photoreceptor maturation, we crossed a floxed allele of *Ctnnb1*, encoding for β-catenin, with a *Pax6-Cre* driver, which we previously confirmed leads to β-catenin loss in P14 *Ctnnb1*-cKO retinas (*Touahri et al., 2024*). P14 wild-type and *Ctnnb1*-cKO retinal sections were immunolabeled with rhodopsin, showing a clear reduction in OS area upon the inhibition of Wnt signaling (*Figure 7D*).

Finally, in chick neuromesodermal progenitors, high glycolytic activity or elevated pHi promotes the non-enzymatic acetylation of β-catenin, which promotes mesodermal at the expense of neural cell fate selection (*Oginuma et al., 2020*). To force protein acetylation, including of β-catenin, cells can be grown in sodium acetate. We observed that the exposure of retinal explants to 5-, 10-, or 20-mM sodium acetate increased the percentage of BrdU$^+$ proliferating RPCs (*Figure 7E*). Thus, the forced activation of β-catenin acetylation is sufficient to promote RPC proliferation, mimicking the early impact of *Pten* loss/high glycolytic activity in RPCs (*Tachibana et al., 2016*). Taken together, our study identifies an unexplored pathway in which PTEN controls RPC proliferation and rod differentiation and maturation via modulating glycolysis (*Figure 7F*). Glycolysis in turn controls proliferation and differentiation through modulating pHi that creates a favorable environment for Wnt signaling activation (*Figure 7F*).

## Discussion

The timing of RPC differentiation and neural subtype specification must be precisely coordinated so that the correct types of retinal neurons and glial cells are born at appropriate times and in their proper numbers during development. In *Pten*-cKO retinas, RPCs proliferate and differentiate on an accelerated timeline (*Jo et al., 2012*; *Sakagami et al., 2012*; *Tachibana et al., 2016*), resulting in the precocious depletion of proliferating RPCs (this study). Ultimately, the changes in RPC proliferation dynamics culminate in the production of fewer amacrine cells and rod photoreceptors populating *Pten*-cKO retinas at P7 and P21 (*Hanna et al., 2025*; *Tachibana et al., 2016*; *Touahri et al., 2024*). To understand how *Pten* controls RPC fate decisions, we first used a transcriptomic approach, revealing an increase in glycolytic gene expression and activity in *Pten*-cKO retinas. Consistent with a role for glycolysis in controlling RPC proliferation and photoreceptor cell differentiation, both were inhibited by 2DG, a pharmacological inhibitor of glycolysis. Conversely, the promotion of glycolytic flux in RPCs using cytoPFKFB3 GOF transgenic mice accelerated photoreceptor differentiation, including the enhanced maturation of rod OSs, an effect phenocopied in *Pten*-cKO retinas. Since elevated glycolytic activity shuttles more lactate/H$^+$ out of the cell to increase pHi, we cultured retinal explants in media buffered to higher pH levels, which promoted RPC proliferation and accelerated photoreceptor differentiation. Finally, we linked glycolysis and pHi changes to the activation of Wnt signaling, which we found is required to promote RPC proliferation, photoreceptor differentiation, and OS maturation. Taken together, we have uncovered a novel molecular axis controlling RPC fate decisions in the retina that begins with *Pten*, a signal transduction molecule, the loss of which elevates glycolytic activity, pHi, and Wnt signaling (*Figure 7F*).

## The role of glycolysis in retinal development across different species

The increased glycolytic gene expression in *Pten*-cKO retinas is likely tied to the increased expression of hypoxia-induced-factor-1-alpha (*Hif1a*), a known target of mTOR signaling and a TF that induces *Slc1a3* (GLUT1) and glycolytic gene expression (*Hanna et al., 2022*). Indeed, mTOR signaling is hyperactive in *Pten*-cKO retinas (*Cantrup et al., 2012*; *Tachibana et al., 2016*; *Tachibana et al., 2018*; *Touahri et al., 2024*), and likewise, in *Tsc1*-cKO retinas, which also increase glycolysis via HIF-1A (*Lim et al., 2021*). Similar to our findings in mice, glycolysis is the predominant mode of ATP production in proliferating *Xenopus* and zebrafish RPCs (*Agathocleous et al., 2012*). However, in contrast to the results of our study, treatment of *Xenopus* retinas with 2DG, a glucose inhibitor, does not impact ATP or lactate production, whereas GPI-mediated inhibition of access to glycogen stores reduces energy production (*Agathocleous et al., 2012*). Thus, *Xenopus* RPCs rely on glycogen storage to fuel glycolysis, whereas we found that in mouse retina, a constant glucose supply is required to fuel glycolysis and support RPC proliferation. Consistent with our data, in mice carrying mutations in *Glut1*, a glucose transporter, or *Ldha*, a glycolytic enzyme that converts pyruvate to lactate, RPC proliferation and expansion of the optic vesicle are inhibited (*Takata et al., 2023*). Human RPCs also use glycolysis as revealed by the conversion of $^{13}$C-glucose to lactate, a glycolytic intermediate (*Takata et al., 2023*). Moreover, the inhibition of glucose catabolism by 2DG blocked RPC proliferation and eye morphogenesis in human retinal organoids (*Takata et al., 2023*). Intriguingly, in this organoid study, RPC identity markers, including RAX and SIX3, were also suppressed by 2DG, a phenotype attributed to lactate depletion, which indicates that glycolysis is also critical for progenitor cell identity (*Takata et al., 2023*). One possibility is that the 'energy budget' of a cell, dictated by ATP and precursor metabolite levels, controls levels of transcription of retinal cell identity genes (*Carthew, 2021*). Glycolytic metabolites can also impact gene transcription in other ways. For example, lactate, a metabolic byproduct of glycolysis, inhibits histone deacetylase (HDAC) activity to acetylate H3K27ac and activate eye development genes in retinal organoids (*Takata et al., 2023*). Indeed, the addition of lactate to retinal organoids induces broad transcriptional changes, including an upregulation of genes involved in eye morphogenesis (*Takata et al., 2023*). Mechanistically, lactate effects on the transcription of eye field markers are attributed to the inhibition of HDACs, which normally inhibit gene expression via histone deacetylation (*Takata et al., 2023*). In the future, we can further assess the impact of 2DG and other metabolic inhibitors on RPC differentiation, focusing on additional retinal cell types, such as bipolar, horizontal, and amacrine cell interneurons, and Müller glia.

## The regulation and role of pHi and Wnt signaling in cell fate decisions

A proxy measure for glycolytic activity are the levels of sentinel metabolites that serve as key indicators of metabolic state (*Kochanowski et al., 2013*; *Miyazawa et al., 2022*; *Oginuma et al., 2020*; *Peeters et al., 2017*; *Zhang et al., 2017*). In chick embryonic tailbuds (*Oginuma et al., 2020*) and the murine presomitic mesoderm (*Miyazawa et al., 2022*), FBP was identified as a sentinel metabolite since its levels positively correlate with glycolytic activity. However, while FBP has been ascribed a major role in controlling mouse embryonic mesoderm development (*Miyazawa et al., 2022*), neuromesodermal fate selection in the chick embryonic tailbud is controlled by elevated pHi induced by high glycolytic activity (*Oginuma et al., 2020*). In the chick tailbud, high pHi creates an environment that activates Wnt signaling through non-enzymatic acetylation of β-catenin (*Oginuma et al., 2020*). Thus, while phosphorylated and non-phosphorylated β-catenin levels do not vary, K49 β-catenin acetylation is reduced at low pHi (*Oginuma et al., 2020*). The onset of K49 β-catenin acetylation is associated with a switch in cell fate choice from neural to mesodermal in the chick tailbud (*Oginuma et al., 2020*). In contrast, in adult cardiomyocytes, when *Pkm*, a rate-limiting glycolytic enzyme, is mutated, β-catenin can more efficiently shuttle to the nucleus to activate Wnt-signaling and promote cardiomyocyte proliferation (*Hauck et al., 2021*). In cardiomyocytes lacking *Pkm*, transcriptional targets of Wnt signaling, including *Axin2*, *Ccnd1*, *Myc*, *Sox2*, and *Tnnt3*, are elevated, and β-catenin protein remains in the cytoplasm, rather than undergoing nuclear translocation (*Hauck et al., 2021*). Similarly, in a study performed in cancer cells, high pH leads to increased expression of *Ccnd1*, a β-catenin target gene, and increases proliferation rates (*Koch et al., 2020*). These findings are consistent with our demonstration that β-catenin levels are stabilized at pH 8, and RPC proliferation is enhanced. Whether pH manipulations also influence cell fate decisions by influencing the stability of other retinal proteins, such as PKM2, can be further investigated in the future using specific

PKM2 inhibitors, such as Shikonin (*Chen et al., 2011*). Finally, in a separate study performed in cancer cells, acidification induced by metformin indirectly suppresses Wnt signaling by activating the DDIT3 transcriptional repressor (*Melnik et al., 2018*), consistent with our data showing low pH suppresses β-catenin stability. *Melnik et al., 2018* also used Mcl inhibitors, as we did in our study, and showed that this treatment blocks Wnt signaling. While we did not examine the impact of CNCn on Wnt signaling, we did observe a decline in proliferation, as expected if Wnt levels are low.

Our finding that Wnt signaling is required to maintain RPC proliferation and drive photoreceptor differentiation is contrary to prior studies that ascribed a role to *Ctnnb1* in maintaining RPCs in an immature, uncommitted state without affecting proliferation (*Ouchi et al., 2011*). A major difference is that *Ouchi et al., 2011* found that *Id3*, a neurogenic inhibitor, is upregulated in *Ctnnb1* cKO retinas, whereas we found that *Id3* expression levels decline in *Pten*-cKO retinas. It may therefore be that the precocious photoreceptor differentiation we observed in the absence of *Pten* occurs due to the reduction in inhibitory *Id3* levels. There may also be a biphasic role for Wnt signaling, as has been observed during neocortical development, where canonical Wnt signaling first induces the symmetric proliferative divisions of neural progenitor cells and later induces differentiation (*Harrison-Uy and Pleasure, 2012*). Finally, the ectopic expression of *Wnt2b* inhibits neuronal differentiation in the chick retina (*Kubo et al., 2005*), phenocopying the postnatal *Pten*-cKO retinal phenotype observed in this study, in which Wnt signaling is also elevated. However, while ectopic *Wnt2b* expression suppresses chick RPC differentiation by inhibiting the expression of proneural genes, we did not observe a significant impact on Notch signaling or proneural gene expression in *Pten*-cKO retinas (data not shown). Thus, the gain of Wnt signaling in *Pten*-cKO retinas impacts RPC proliferation and differentiation possibly in a proneural gene-independent manner.

The link between glycolysis and Wnt signaling may be species and lineage specific. For instance, in mouse embryos, *Miyazawa et al., 2022* found that increasing glycolysis in the presomitic mesoderm inhibits Wnt signaling. Furthermore, in a study performed in Madin–Darby canine kidney cells, higher pH levels were shown to decrease β-catenin levels in the cytoplasm, nucleus, and junctional complexes (*Czowski and White, 2024*). In this study, the authors altered pH using inhibitors for a sodium-proton exchanger and a sodium bicarbonate transporter, in contrast to the *Oginuma et al., 2020* study, which instead used the ionophores nigericin and valinomycin to equilibrate intracellular pHi to media pH. Consistent with the idea that there is some species specificity, in human retinal organoids, lactate itself was shown to alter optic vesicle bud gene expression, whereas pH did not appear to be an important driver of cell fate decisions (*Takata et al., 2023*). In contrast, in the mouse retina, the GPR81 receptor for lactate is expressed at the highest levels in ganglion cells in which it is required to promote axon growth, with only low, possibly background expression levels in RPCs and photoreceptors (*Laroche et al., 2021*). In the future, to more comprehensively examine the link between *Pten* loss, glycolytic activity, pHi and Wnt signaling, we could examine levels of phosphorylated, non-phosphorylated and K49 acetylated β-catenin after each manipulation (i.e., *Pten* loss, pH manipulations, CNCn treatment, glycolysis inhibition, and acetate treatments). By collecting CD133$^+$ MACS-enriched RPCs from the retinal explants, we could add cell type and stage specificity to our analysis. Finally, the use of qPCR to examine Wnt signaling genes, such as *Axin2*, *Ccnd1*, *Myc*, *Sox2*, and *Tnnt3*, would provide further support for the impact of glycolysis and pH on this signaling pathway. Lastly, an important caveat to our study is that metabolism changes ex vivo versus in vivo, and thus, in the future, in vivo studies can be performed to assess metabolic changes.

## Role of *Pten* in photoreceptor differentiation and maturation

Our finding that *Pten* loss drives photoreceptor differentiation and OS maturation was an unexpected fate bias given that during development, *Pten* is expressed in multipotent RPCs (*Tachibana et al., 2016*) that give rise to all retinal cells in a defined temporal sequence (*Javed and Cayouette, 2017*). RPCs that are biased to a generic photoreceptor fate express the TF Olig2; if these cells later express the TF Nrl, they become rods, if not, they become cones (*Hafler et al., 2012*; *Javed and Cayouette, 2017*). In *Pten*-cKO retinas, more RPCs become photoreceptors instead of differentiating into other retinal cell types at early developmental stages (*Tachibana et al., 2016*), but this phenotype is not likely to be dependent on Olig2, since Olig2 is expressed at normal levels from the RNA-seq study performed herein. It is possible that glycolytic inhibition with 2DG slows down the development and production of most newly differentiated cells rather than specifically affecting photoreceptor

differentiation, which we could assess in the future. Nevertheless, the finding that *Pten* loss elicits a photoreceptor bias in early-RPCs could have widespread applications in targeting uncurable retinal diseases. Notably, the pharmacological inhibition of PTEN has been used in pre-clinical studies in lung, muscle, and nervous systems for regenerative purposes (*Borges et al., 2020*; *Knafo and Esteban, 2017*), including in the retina, to promote ganglion cell axonogenesis (*Park et al., 2008*).

In summary, we have revealed that elevated glycolysis is a key contributor to the RPC fate alterations observed in *Pten*-cKO retinas. In particular, we have found that glycolytic flux is elevated in *Pten*-cKO retinas, which modulates both pHi and Wnt signaling to control the timing of RPC proliferation and differentiation.

## Materials and methods
### Mice
All animal procedures were approved by the Sunnybrook Research Institute Animal Care Committee (16-606) in agreement with the Guidelines of the Canadian Council of Animal Care (CCAC). The following transgenic animal lines were used from Jackson Laboratory: *Pten$^{fl}$* allele (*Backman et al., 2001*) (B6.129S4-*Pten$^{tm1Hwu}$*/J. Strain #: 006440, RRID:IMSR_JAX:006440), *Ctnnb1$^{fl}$* (B6(Cg)-Ctnnb1tm1Knw/J. Strain #: 022775, RRID:IMSR_JAX:022775), and *Pax6-Cre* driver (*Marquardt et al., 2001*) (STOCK Tg(Pax6-GFP/cre)1Rilm/J. Strain #:024578. RRID:IMSR_JAX:024578. Common Name: P0-3.9GFPCre). For glycolysis GOF studies, cytoPFKFB3-GOF mice (EMBL, Germany) (*Miyazawa et al., 2022*) were crossed with a *Pax6-Cre* driver to generate cytoPFKFB3-GOF transgenic mice. PCR genotyping was performed as described by Jax for commercial lines and as described for the cytoP-FKFB3 allele (*Miyazawa et al., 2022*). All lines were maintained on a C57BL/6J background through backcrossing (Strain #:000664. RRID:IMSR_JAX:000664. Common Name: B6). Animal explants were performed with CD-1 mice from Charles River (Strain #022, RRID:IMSR_CRL:022). The day of the vaginal plug was considered to be embryonic day E0.5 for timed pregnancies.

### Bulk RNA-seq
P0 *Pten*-cKO and wild-type control eyes were dissected in a clean petri dish containing Dulbecco's phosphate-buffered saline. A pair of super-fine tweezers (#5, Fine Scientific Tools, Canada) was used to remove the cornea, iris, sclera, and RPE. Retinas were homogenized using TRIzol (Thermo Fisher Scientific; #15596-026) according to the manufacturer's protocol. RNA was then purified using an RNAeasy minikit (74104, QIAGEN). Eluted RNA quality and concentration was assessed using a Nano-Drop. Samples with the highest purity (260/280 ratio = 2.0) were used for RNA-seq experiments. Five *Pten*-cKO and four wild-type samples were sent to the University of Lethbridge, Alberta for sequencing. Illumina NextSeq 500 performed sequencing using 75 bp Single-End reads. A NEBNext Ultra II Directional RNA Library Prep Kit was used to construct a library containing read counts and was compared to a Mouse GRCM38 reference genome. DEGs were identified using the DESeq1 1.28.1 package. An adjusted p-value (padj) less than 0.05 for any gene was considered a DEG (Wald test, Benjamini–Hochberg correction for multiple comparisons). DEGs were visualized by Volcano plot (R package: EnhancedVolcano 1.6.0) and heatmap (R package: gplots 3.0.3, function heatmap 2). Gene Ontology terms for Biological Processes and KEGG pathways were assayed using the DEG up- and downregulated DEGs (Pathview 1.28.0 package).

### In vivo injections
For birthdating and proliferation studies, animals were injected with BrdU at 10 mg/kg intraperitoneally at E12.5 and then harvested at P7. For 2DG in vivo experiments, newborn pups were injected with 2DG intraperitoneally (30 mg/kg, Sigma-Aldrich, Oakville, Canada) daily starting at P0 until P7.

### Tissue processing and immunostaining
Animals were sacrificed using $CO_2$ euthanasia. Eyes were dissected and fixed in 4% paraformaldehyde (PFA) overnight at 4°C. Fixed eyes were rinsed with PBS and incubated in 20% sucrose overnight. Eyes were then embedded in optical cutting temperature (OCT) compound (Tissue-Tek, Sakura Finetek U.S.A. Inc, Torrance, CA) and frozen on dry ice before being sectioned in a series of 10 µM slices using a Leica CM3050s cryostat (Leica Biosystems, Buffalo Grove, IL, USA). Sections

were mounted on Fisherbrand Superfrost Plus slides (Thermo Fisher Scientific, Markham, ON). Sections were blocked for 1 hr in 10% normal horse serum in 1× PBS/0.1% Triton X-100 (PBST). Primary antibodies were diluted in a blocking solution and incubated at 4°C overnight. Primary antibodies included: BrdU (1:400, ab6326, Abcam), VSX2 (1:200, sc-365519, Santa Cruz), Ki67 (1:400, ab16667, Abcam), Cleaved Caspase 3 (CC3) (1:400, ab2302, Abcam), Rhodopsin (RHO) (1:400, MAB 5356, Millipore), PAX6 (1:400, #PRB-278P; Covance Research), PTEN (1:400, 9559S, Cell Signaling) and CRX (1:400, NBP2-15964, Novus Biologicals). Slides were washed 3 × 10 min in PBST and then incubated in secondary antibodies diluted in PBST for 1 hr. Secondary antibodies were conjugated to Alexa568 (1:500, Molecular Probes), Alexa488 (1:500, Molecular Probes), or Alexa647 (1:500, Molecular Probes). Sections were washed 3 × 10 min washes in PBST. Sections were counterstained with 4′,6-diamidino-2-phenylindole (Invitrogen, D1306) and coverslipped in Aqua-Poly/Mount (Polysciences #18606).

## Retinal explant assay

For retinal explants, P0 wild-type CD1 mice pups were sacrificed by decapitation. Eyes were dissected in ice-cold PBS. Corneas were removed first, followed by removal of the sclera, iris, lens, and RPE. Retinal cups were then mounted on Cytiva Whatman 90 mm Nucleopore Polycarbonate Track-Etched Membranes (Catalog Number 09920022, Cytiva, Canada). Four cuts were made to flatten cups on the membranes using TMS101 Gills-Vannas Scissors (SKU: TMS101, Titan Medical, Coral Gables, FL, USA). Retinas mounted on Nucleopore membranes were then transferred to 24-well plates (Corning Costar TC-Treated Multiple Well Plates, Catalog Number: CLS3524, Millipore Sigma) filled with 1 ml of retinal explant media: 50% DMEM-high glucose (25 mM), no glutamine, no methionine, no cystine (Gibco, REF 21013-024), 25% HBSS (Cat 311-512-CL), 25% heat-inactivated horse serum (Wisent, Cat 065-150), 200 µM L-glutamine (Wisent, Cat 609-065-EL), 0.6 mm HEPES (Wisent, Cat 330-050-CL), and 1% Pen-Strep (Wisent, Cat 450-201-EL). Retinas were incubated at 37°C, 5% $CO_2$. At the experimental endpoint (as indicated), explants were fixed at room temperature with 4% PFA for 15 min, rinsed with DEPC-treated PBS, and incubated in 20% DEPC-Sucrose overnight. Explants were then mounted in OCT on dry ice and cryosectioned at 10 µM as described above.

## Retinal explant drug treatments

For BrdU labeling, 5 µl of 10 mg/ml BrdU was added to the retinal cell explant media for the time period described. All inhibitors were incubated with retinal explants for 24 hr. 2DG (D3179-1G, Sigma-Aldrich, Canada) was diluted in PBS and added to retinal explant media to achieve 5 and 10 mM final concentrations. 10 mM 2DG was previously shown to reduce lactate levels in tumor cells (*Malm et al., 2015*). We confirmed the efficacy of our 2DG batch on reducing lactate levels in the retina when administered in vivo (*Hanna et al., 2025*). GPI (361515-1MG, Sigma-Aldrich, Canada) was diluted in DMSO and added to the retinal explant media at a final concentration of 12.5 or 25 µM for 24 hr. α-Cyano-4-hydroxy-cannamic acid (CNCn; C2020-10G; Sigma-Aldrich, Canada) was diluted in explant media and added to explant media at a final concentration of 5 mM for 24 hr. For pH experiments, pH was adjusted after incubating the media in a cell culture incubator for 1 hr using hydrochloric acid or sodium hydroxide to reach a pH of 6.5 and 8.0, respectively, and grown for 24 hr. FH535 was diluted in DMSO and added to the retinal explant media at 5 and 10 mM concentrations for 24 hr (108409, Millipore Sigma, Canada). For acetate explants, retinas were incubated with sodium acetate at a concentration of 5, 10, or 20 mM diluted in PBS for 48 hr (S2889, Millipore Sigma, Canada).

## Retinal explant electroporation

P2 wild-type retinal explants were electroporated with pCAGG-pHluorin-IRES2-tdTomato obtained from Olivier Pourquié (*Oginuma et al., 2020*) as described (*Tachibana et al., 2016*). Briefly, retinas were placed in 2 mm gap cuvettes (VWR International) with 30 µl of DNA at 0.5 µg/µl. Retinas were then electroporated using an ECM830 pulse generator (BTX Harvard Apparatus) using five square 20 V pulses of 50-ms duration, separated by 950-ms intervals. After electroporation, retinas were cultured on Nucleopore Polycarbonate Track-Etched Membranes in retinal explant media at 37°C, 5% $CO_2$.

## qPCR

TRIzol RNA Isolation Reagent (Thermo Fisher Scientific; #15596-026) was used to extract RNA according to the manufacturer's instructions. cDNA was synthesized from 0.5 µg RNA using an RT² first-strand kit (QIAGEN, #330401) according to the manufacturer's instructions. qPCR was performed using an RT² SYBR Green PCR Master Mix (QIAGEN; #330500) as described by the manufacturer, with a CFX384 cycler (Bio-Rad Laboratories, Canada). The following primers were used: *Eno1* (PPM04408B, QIAGEN), *Hk1* (PPM05501A, QIAGEN), *Pgk1* (PPM03700A, QIAGEN), and *Slc16a3* (PPM27328A, QIAGEN) and normalized to housekeeping genes *Hprt* (PPM03559F, QIAGEN), *B2m* (PPM03562A, QIAGEN), and *Mapk1* (PPM03571E, QIAGEN).

## Western blotting

For western blot, retinal cups were dissected on ice and both retinas of the same animal were pooled together. Retinas were lysed in NP-40 lysis buffer (0.05 M Tris, pH 7.5, 0.15 M NaCl, 1% NP-40, 0.001 M EDTA) with protease (1× protease inhibitor complete, 1 mM PMSF), proteasome (MG132 at 0.05 mM) and phosphatase (50 mM NaF, 1 mM NaOV) inhibitors. Protein amounts were quantified with a Micro BCA Protein Assay Kit (Thermo Fisher Scientific, #23235) following the manufacturer's instructions. 10 µg of lysate from whole cells or tissue was run on 10% SDS–PAGE gels for western blot analysis. Protein was transferred to PVDF membrane at 80 V for 1 hr. PVDF membranes were then blocked in 5% skim milk powder in tris-buffered saline with 0.1% tween 20 (TBST) for 1 hr at room temperature. Blots were incubated overnight at 4°C in a blocking solution with non-phosphorylated β-catenin (1:1000, 8814, Cell Signaling) followed by β-actin (1:5000, ab8227, Abcam). Blots were washed 3 × 10 min in TBST before incubating in secondary antibodies, including either Goat Anti-Rabbit IgG (H+L)-HRP Conjugate (1/50,000; Bio-Rad #1721019 or Cell Signaling #7074) or Goat Anti-Mouse IgG (H+L)-HRP Conjugate (1/50,000; Bio-Rad #1721011). A chemiluminescent signal was obtained from the blots using (ECL Prime Luminor Enhancer Solution, Cytiva, Canada) and the blot was developed using a ChemiDoc System (Bio-Rad Laboratories, Canada). Densitometries were calculated using ImageJ (*Schneider et al., 2012*), and normalized values (over β-actin) were plotted.

## scRNA-seq analysis

10× scRNA-seq data of mouse developing retina were obtained from GSE118614 (*Clark et al., 2019*). Further processing and analyses were performed with the Seurat v.3.2.3 R package (*Butler et al., 2018*). Low-quality cells were excluded by filtering out cells with fewer than 500 detected genes and cells with mitochondrial RNA more than 5%. The data were then transformed by the SCTransform function while regressing out the variance due to mitochondrial RNAs. Clustering was performed by the RunPCA, FindNeighbors, and FindClusters functions using the first 30 principal components. The 2D projection of the clustering was carried out by the RunUMAP function. The annotation of cell type to each cluster was performed by using the same set of markers as in *Clark et al., 2019*. Expression of selected genes was plotted using the FeaturePlot function.

## Seahorse assay

P0 *Pten*-cKO and wild-type control pups were sacrificed, and retinal cups were dissected. Both retinas from the same animal were pooled together in one tube and dissociated using a Papain kit according to the manufacture's protocol (Papain Dissociation System, Worthington Biochemical Corporation, USA). Dissociated cells of each animal were then plated in a 6-well plate coated with poly-D-lysine (10 mg/ml) (Corning, 354210)/Laminin (Corning, 354232) (2.5 µg/ml) (Sarstedt, Germany). Cells were cultured for 5 days then transferred to Seahorse 96-well plate at a density of 40,000/well. A Seahorse Mito stress test was run according to the manufacture's protocol. Seahorse basal media was supplemented with oligomycin, FCCP, R/A at the indicated time points (Seahorse XF Mito Stress Test Kit, 103015-100, Agilent, USA). Readings were then normalized to protein concentration/well measured using the Micro BCA Protein Assay Kit.

## Lactate assay

To measure lactate levels, P0 wild-type, *Pten*-cKO, or cytoPFKFB3-GOF retinas were dissected on ice-cold PBS, and both retinas of each pup were pooled together in the same tube. Cell lysates were prepared as described for using NP-40 lysis buffer and protein levels were quantified using the

Micro BCA Protein Assay Kit. Measurements of lactate were then performed using a high sensitivity Lactate Colorimetric/Fluorometric Assay Kit as per the manufacturer's protocol (K607-100; BioVision; CA, USA) and normalized to protein levels.

### Imaging, quantification, and figure preparation

Retinal cross sections were imaged using a Leica DM18 inverted fluorescent microscope and Leica Application Suite X (LASX) software for cell counts (Leica Biosystems, Canada) or a Nikon A1 laser scanning confocal microscope at Sunnybrook Research Institute Centre for Cytometry and Scanning Microscopy (Nikon, Canada). OS measurements were performed by imaging three different retinal cross sections stained with rhodopsin ($n = 3$) of each of three different biological replicates ($N = 3$) for each animal group. OSs were then outlined using ImageJ and the total area was calculated. Licenses for Adobe Creative Commons (Photoshop and Illustrator) 2021 were used to prepare figures and schematics.

### Statistical analysis

Statistical analysis was performed on at least three different animals/explants ($N$; biological replicates). Cell counts were performed by counting cells in three different cross sections ($n$; technical replicates). For comparisons between two groups, an unpaired two-tailed Student's $t$-test was used (GraphPad Prism, USA). For comparisons involving more than two groups, one-way ANOVA with post hoc Tukey multiple comparisons of means was used (GraphPad Prism, USA). Results were plotted on graphs with SEM error bars (GraphPad Prism, USA). A p-value less than 0.05 was considered statistically significant.

## Acknowledgements

We would like to thank Alexander Aulehla for providing the cytoPFKFB3 transgenic mice. We thank Olivier Pourquié for providing reagents, Robert Cantrup in the Schuurmans' lab and Petia Stefanova in the SRI Histology Facility for help with sectioning and immunostaining, and Cathy Ioria in RS lab for help with the Seahorse assay. This work was supported by the Canadian Institutes of Health Research (CIHR) PJT 180243 to CS, RS, YT and IK, and PJT – 400973 to IA and CS. JH was supported by a Canada Graduate Scholarship – Doctoral (CGS-D)/CIHR award, Vision Science Research Program Scholarship, Ontario Graduate Scholarship, and R.O. Torrance Bursary. CS holds the Dixon Family Chair in Ophthalmology Research at the Sunnybrook Research Institute.

## Additional information

### Funding

| Funder | Grant reference number | Author |
|---|---|---|
| Canadian Institutes of Health Research | PJT 180243 | Carol Schuurmans Robert A Screaton Yacine Touahri Igor Kovalchuk |
| Canadian Institutes of Health Research | PJT – 400973 | Isabelle Aubert Carol Schuurmans |
| Government of Canada | Canada Graduate Scholarship – Doctoral | Joseph Hanna |
| University of Toronto | Vision Science Research Program Scholarship | Joseph Hanna |
| Government of Ontario | Ontario Graduate Scholarship | Joseph Hanna |
| University of Toronto | R.O. Torrance Bursary | Joseph Hanna |

The funders had no role in study design, data collection, and interpretation, or the decision to submit the work for publication.

## Author contributions
Joseph Hanna, Conceptualization, Investigation, Visualization, Methodology, Writing – original draft, Writing – review and editing; Yacine Touahri, Investigation, Visualization, Methodology, Writing – original draft, Writing – review and editing; Alissa Pak, Luke Ajay David, Sisu Han, Investigation, Visualization, Methodology, Writing – review and editing; Lauren Belfiore, Visualization, Methodology, Writing – review and editing; Edwin van Oosten, Yaroslav Ilnytskyy, Satoshi Okawa, Investigation, Methodology, Writing – review and editing; Igor Kovalchuk, Antonio del Sol, Supervision, Methodology, Writing – review and editing; Deborah Kurrasch, Supervision, Writing – review and editing; Robert A Screaton, Supervision, Funding acquisition, Methodology, Writing – review and editing; Isabelle Aubert, Supervision, Funding acquisition, Writing – review and editing; Carol Schuurmans, Conceptualization, Supervision, Funding acquisition, Methodology, Writing – original draft, Project administration, Writing – review and editing

## Author ORCIDs
Alissa Pak ⓘ https://orcid.org/0009-0001-4981-1950
Deborah Kurrasch ⓘ https://orcid.org/0000-0002-9945-287X
Antonio del Sol ⓘ https://orcid.org/0000-0002-9926-617X
Carol Schuurmans ⓘ https://orcid.org/0000-0003-3567-0058

## Ethics
All animal procedures were approved by the Sunnybrook Research Institute Animal Care Committee (16-606) in agreement with the Guidelines of the Canadian Council of Animal Care (CCAC).

Reviewer #1 (Public review): https://doi.org/10.7554/eLife.100604.3.sa1
Reviewer #3 (Public review): https://doi.org/10.7554/eLife.100604.3.sa2
Author response https://doi.org/10.7554/eLife.100604.3.sa3

---

# Additional files

## Supplementary files
MDAR checklist

## Data availability
The primary bulk RNA-sequencing data from this study are not available due to a corrupted hard drive that occurred during the course of peer review, precluding us from uploading our raw data. Instead, the preprocessed data file is available with RPKM values for each gene (Figure 2—source data 1). The mined scRNA-seq data from the developing mouse retina were obtained from GSE118614 (*Clark et al., 2019*). Uncropped western blots have been deposited in Mendeley (https://data.mendeley.com/datasets/r4t3vv372f/1). The source data files have been provided for Figure 6.

The following dataset was generated:

| Author(s) | Year | Dataset title | Dataset URL | Database and Identifier |
|---|---|---|---|---|
| Touahri Y, Schuurmans C | 2025 | B Catenin is a downstream mediator of Pten and glycolysis | https://doi.org/10.17632/r4t3vv372f.1 | Mendeley Data, 10.17632/r4t3vv372f.1 |

The following previously published dataset was used:

| Author(s) | Year | Dataset title | Dataset URL | Database and Identifier |
|---|---|---|---|---|
| Clark BS, Stein-O'Brien GL, Shiau F, Cannon GH, Davis-Marcisak E, Sherman T, Santiago CP, Hoang TV, Rajaii F, James-Esposito RE, Gronostajski RM, Fertig EJ, Goff LA, Blackshaw S | 2019 | Single-cell RNA-Seq Analysis of Retinal Development Identifies NFI Factors as Regulating Mitotic Exit and Late-Born Cell Specification | https://www.ncbi.nlm.nih.gov/geo/query/acc.cgi?acc=GSE118614 | NCBI Gene Expression Omnibus, GSE118614 |

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
