## [Editor Report · eLife Assessment]

This **fundamental** study advances our understanding of the role that energy metabolism, specifically anaerobic glycolysis, plays during retinal development. **Convincing** in vitro genetic and pharmacological evidence demonstrates that glycolytic flux controls retinal progenitor cell proliferation rates and the timing of photoreceptor maturation. Interesting evidence suggests potential downstream roles for intracellular pH and Wnt/β-catenin signaling; however, more direct evidence is needed to show they are the key mechanisms through which glycolytic flux regulates retinogenesis in vivo. This work is expected to stimulate broad interest and possible future studies investigating the link between metabolism and development in other tissue systems.

[Editors’ note: Primary data for this manuscript are not available due to a corrupted hard drive that occurred during the course of peer review. However, preprocessed data are available.]

---

## [Referee Report · Reviewer #1 (Public review)]

Summary:

This paper seeks to understand the upstream regulation and downstream effectors of glycolysis in retinal progenitor cells, using mouse retinal explants as the main model system. The paper presents evidence that high glycolysis in retinal progenitor cells is required for their proliferation and timely differentiation into photoreceptors. Retinal glycolysis increases after deletion of Pten. The authors suggest that high glycolysis controls cell proliferation and differentiation by promoting intracellular alkalinization, beta-catenin acetylation and stabilization and consequent activation of the canonical Wnt pathway.

Strengths:

- The experiments showing that PFKFB3 overexpression is sufficient to increase proliferation of retinal progenitors (which are already highly dividing cells) and photoreceptor differentiation are striking and the result unanticipated. It suggests that glycolytic flux is normally limiting for proliferation in embryos.

- Likewise the result that an increase in pH from 7.4 to 8.0 is sufficient to increase proliferation implies that pH regulation may have instructive roles in setting the tempo of retinal development and embryonic cell proliferation. Similarly for the results showing that acetate supplementation increases proliferation (I think this result should be moved to the main figures).

Weaknesses:

- Epistatic experiments to test if changes in pH mediate the effects of glycolysis on photoreceptor differentiation, or if Wnt activation is the main downstream effector of glycolysis in controlling differentiation are not presented.

- It is likely that metabolism changes ex vivo vs in vivo, and therefore stable isotope tracing experiments in the explants may not reflect in vivo metabolism.

- The retina at P0 is composed of both progenitors and differentiated cells. It is not clear if the results of the RNA-seq and metabolic analysis reflect changes in the metabolism of progenitors, or of mature cells, or changes in cell type composition rather than direct metabolic changes in a specific cell type.

- The biochemical links between elevated glycolysis and pH and beta-catenin stability are unclear. White et al found that higher pH decreased beta-catenin stability (JCB 217: 3965) in contrast to the results here. Oginuma et al found that inhibition of glycolysis or beta-catenin acetylation does not affect beta-catenin stability (Nature 584:98), again in contrast to these results. Another paper showed that acidification inhibits Wnt signaling by promoting the expression of a transcriptional repressor and not via beta-catenin stability (Cell Discovery 4:37). There are also additional papers showing increased pH can promote cell proliferation via other mechanisms (e.g. Nat Metab 2:1212). It is possible that there is organ-specificity in these signaling pathways however some clarification of these divergent results is warranted.

- The gene expression analysis is not completely convincing. E.g. expression of additional glycolytic genes should be shown in Fig. 1. It is not clear why Hk1 and Pgk1 are specifically shown, and conclusions about changes in glycolysis are difficult to draw from expression of these two genes. The increase in glycolytic gene expression in the Pten-deficient retina is generally small.

- Is it possible that glycolytic inhibition with 2DG slows down development and production of most new differentiated cells rather than specifically affecting photoreceptor differentiation?

- Are the prematurely-born cells caused by PFKFB3 overexpression photoreceptors as assessed by morphology or markers (in addition to position)?

---

## [Referee Report · Reviewer #3 (Public review)]

Summary:

This study examines the metabolic regulation of progenitor proliferation and differentiation in the developing retina. The authors observe dynamic changes in glycolytic gene expression in retinal progenitors and use various strategies to test the role of glycolysis. They find that elevated glycolysis in Pten-cKO retinas results in alteration of RPC fate, while inhibition of glycolysis has converse effects. They specifically test the role of elevated glycolysis using dominant active cytoPFKFB3, which demonstrates the selective effects of elevated glycolysis on progenitor proliferation and rod differentiation. They then show that elevated glycolysis modulates both pHi and Wnt signaling, and provide evidence that these pathways impact proliferation and differentiation of progenitors, particularly affecting rod photoreceptor differentiation.

Strengths:

This is a compelling and rigorous study that provides an important advance in our understanding of metabolic regulation of retina development, addressing a major gap in knowledge. A key strength is that the study utilizes multiple genetic and pharmacological approaches to address how both increased or decreased glycolytic flux affect retinal progenitor proliferation and differentiation. They discover elevated Wnt signaling pathway genes in Pten cKO retina, revealing a potential link between glycolysis and Wnt pathway activation. Altogether the study is comprehensive and adds to the growing body of evidence that regulation of glycolysis plays a key role in tissue development.

Weaknesses:

(1) Following expression of cytoPFKFB3, which results in increased glycolytic flux, BrDU labeling was performed at e12.5 and increased labeled cells were detected in the outer nuclear layer. But whether these are cones or rods is not established. The rest of the analysis is focused on the precocious maturation of rhodopsin-labelled outer segments, and the major conclusions emphasize rod photoreceptor differentiation. Therefore it is unclear whether there is an effect on cone differentiation for either Pten cKO or cytoPFKFB3 transgenic retina. It is also not established whether rods are born precociously. Presumably this would be best detected by BrDU labeling at later embryonic stages.

(2) The authors find that there is upregulation of multiple Wnt pathway components in Pten cKO retina. They further show that inhibiting Wnt signaling phenocopies the effects of reducing glycolysis. However, they do not test whether pharmacological inhibition of Wnt signaling reverses the effects of high glycolytic activity in Pten cKO retinas. Thus the argument that Wnt is a key downstream effector pathway regulating rod photoreceptor differentiation is weak.

(3) The use of sodium acetate to force protein acetylation is quite non-specific and will have effects beyond beta-catenin acetylation (which the authors acknowledge). Thus it is a stretch to state that "forced activation of beta-catenin acetylation" mimics the impact of Pten

loss/high glycolytic activity in RPCs since the effects could be due to acetylation of other proteins.

---

## [Author Response]

The following is the authors’ response to the original reviews

**Reviewer 1 (Public review):**
(1) “It is likely that metabolism changes ex vivo vs in vivo, and therefore stable isotope tracing experiments in the explants may not reflect in vivo metabolism.”

We agree with the reviewer that metabolic changes may differ ex vivo versus in vivo. We now state: “Lastly, an important caveat to our study is that metabolism changes ex vivo versus in vivo, and thus, in the future, in vivo studies can be performed to assess metabolic changes.” (lines 591-593).

(2) “The retina at P0 is composed of both progenitors and differentiated cells. It is not clear if the results of the RNA-seq and metabolic analysis reflect changes in the metabolism of progenitors, or of mature cells, or changes in cell type composition rather than direct metabolic changes in a specific cell type.”

We have clarified that the metabolic changes may be in RPCs or in other retinal cell types on lines 149-152: “Since these measurements were performed in bulk, and the ratio of RPCs to differentiated cells declines as development proceeds, it is not clear whether glycolytic activity is temporally regulated within RPCs or in other retinal cell types.”

However, since we mined a single cell (sc) RNA-seq dataset, we are able to attribute gene expression specifically within RPCs (Figure 1).

(3) “The biochemical links between elevated glycolysis and pH and beta-catenin stability are unclear. White et al found that higher pH decreased beta-catenin stability (JCB 217: 3965) in contrast to the results here. Oginuma et al found that inhibition of glycolysis or beta-catenin acetylation does not affect beta-catenin stability (Nature 584:98), again in contrast to these results. Another paper showed that acidification inhibits Wnt signaling by promoting the expression of a transcriptional repressor and not via beta-catenin stability (Cell Discovery 4:37). There are also additional papers showing increased pH can promote cell proliferation via other mechanisms (e.g. Nat Metab 2:1212). It is possible that there is organ-specificity in these signaling pathways however some clarification of these divergent results is warranted.”

We have added the information and references brought up by the reviewer in our discussion (lines 529-549 and 570-574). We have also suggested future experiments to further analyse our system in line with the studies now referenced (lines 580-589).

(4) The gene expression analysis is not completely convincing. E.g. the expression of additional glycolytic genes should be shown in Figure 1. It is not clear why Hk1 and Pgk1 are specifically shown, and conclusions about changes in glycolysis are difficult to draw from the expression of these two genes. The increase in glycolytic gene expression in the Pten-deficient retina is generally small.

We have expanded the list of glycolytic genes analysed, in modified Figure 1B, and expanded the description of these results on lines 156-166.

(5) Is it possible that glycolytic inhibition with 2DG slows down the development and production of most newly differentiated cells rather than specifically affecting photoreceptor differentiation?

We added a comment to this effect to the discussion: “It is possible that glycolytic inhibition with 2DG slows down the development and production of most newly differentiated cells rather than specifically affecting photoreceptor differentiation, which we could assess in the future.“ (lines 600-603).

(6) “Likewise the result that an increase in pH from 7.4 to 8.0 is sufficient to increase proliferation implies that pH regulation may have instructive roles in setting the tempo of retinal development and embryonic cell proliferation. Similarly, the results show that acetate supplementation increases proliferation (I think this result should be moved to the main figures).”

We have added the acetate data to main Figure 7E.

We added a supplemental data table that was inadvertently not included in our last submission. Figure 2– Data supplement 1.

**Reviewer #2 (Recommendations for the authors):**
Major points(1) Assuming that increased glycolysis gets RPCs to exit from the proliferative stage earlier, the total number of retinal cells, notably that of the rod photoreceptors, should be reduced since the pool of proliferating cells is depleted earlier. Is that really the case for a mature retina? To address this question, the authors should perform quantifications of photoreceptors at a stage where most developmental cell death has concluded (i.e. at P14 or later; Young, J. Comp. Neurol. 229:362-373, 1984) and check whether or not there are more or less photoreceptors present.

We have previously quantified numbers of each cell type in *Pten* RPC-cKO retinas, and as suggested by the reviewer, there are fewer rod photoreceptors at P7 (Tachibana et al. 2016. J Neurosci 36 (36) 9454-9471) and P21 (Hanna et al. 2025. IOVS. Mar 3;66(3):45). We have edited the following sentence: “Using cellular birthdating, we previously showed that *Pten*-cKO RPCs are hyperproliferative and differentiate on an accelerated schedule between E12.5 and E18.5, yet fewer rod photoreceptors are ultimately present in P7 (Tachibana et al., 2016) and P21 (Hanna et al., 2025) retinas, suggestive of a developmental defect. (lines 184-187).

(2) Figure 1B, 1H: On what data are these two figures based? The plots suggest that a high-density time series of gene expression and rod photoreceptor birth was performed, yet it is not clear where and how this was done. The authors should provide the data, plot individual data points, and, if applicable perform a statistical analysis to support their idea that glycolytic gene expression (as a surrogate for glycolysis) overlaps in time with rod photoreceptor birth (Figure 1B) and that in Pten KO the glycolytic gene expression is shifted forward in time (Figure 1H). If the data required to construct these plots (min. 5 data points, min 3 repeats each) does not exist or cannot be generated (e.g. from reanalysis of previously published datasets), then these graphs should be removed.

We have removed the previous Figure 1B and Figure 1H.

(3) Figure 2E: Which PKM isozyme was analyzed here? Does the genetic analysis allow us to distinguish between PKM1 and PKM2? Since PKM governs the key rate-limiting step of glycolysis but was not significantly upregulated, does this not contradict the authors' main hypothesis? If PKM at some point was inhibited (see also below comment to Figure 5) one would expect an accumulation of glycolytic intermediates, including phosphoenolpyruvate. Was such an effect observed?

The data in Figure 2E is bulk RNA-seq data. Since there is only a single *Pkm* gene that is alternatively spliced, the RNA-sequencing data cannot distinguish between the four PK isozymes that arise from alternative splicing. Specifically, we used Illumina NextSeq 500 for sequencing of 75bp Single-End reads that will sequence transcripts for alternatively spliced *Pkm1* and *Pkm2* mRNAs, which carry a common 3’end. We added a statement to this effect: “However, since we employed 75 bp single-end sequencing, we could not distinguish between alternatively spliced *Pkm1* and *Pkm2* mRNAs.“ (lines 215-216).

We have not performed metabolic analyses of glycolytic intermediates, but we have proposed such a strategy as an important avenue of investigation for future studies in the Discussion: “Lastly, an important caveat to our study is that metabolism changes ex vivo versus in vivo, and thus, in the future, in vivo studies can be performed to assess metabolic changes.” (lines 591-593).

(4) Figure 3 and materials & methods: For the retinal explant cultures, was the RPE included in the cultured explants? If so, how can the authors distinguish drug effects on neuroretina and RPE? If the RPE was not included, then the authors should discuss how the missing RPE - neuroretina interaction could have influenced their results.

We remove the RPE from the retinal explants, as indicated in the Methods section. The RPE is a metabolic hub that allows transport of nutrients for the retina, so in the absence of the RPE, there is not an immediate source of energy, such as glucose, to the retina. However, the media (DMEM) contains 25 mM glucose to replace the RPE as an energy source, and we now show that RPCs express GLUT1, which allows uptake of glucose (see new Figure 3A).

We added the following sentence “P0 explants were mounted on Nucleopore membranes and cultured on top of retinal explant media, providing a source of nutrients, growth factors and glucose. “(lines 241-243).

(5) Figure 3: It seems rather odd that, if glycolysis was so important for retinal proliferation, differentiation, and metabolism in general, the inhibition of glycolysis with 2DG should not produce a strong degeneration. However, since 2DG competes with glucose, and must be used at nearly equimolar concentration to block glycolysis in a meaningful way, it is possible that the 2DG concentration used simply was not high enough to substantially inhibit glycolysis. Since the inhibitory effect of 2DG depends on the glucose concentration, the authors should measure and provide the concentration of glucose in the explant culture medium. This value should be given either in results or materials and methods.

We recently published a manuscript showing that 2DG treatments at the same concentrations employed in this study are effective at reducing lactate production in the developing retina in vivo, which is the expected effect of reduced glycolysis (Hanna et al. 2025. IOVS). However, in this study, we did not observe an impact on cell survival.

We do not agree that it is necessary to measure glucose in the media since the anti-proliferative effect of 2DG is well known, and we are working in the effective range established by multiple groups. We have clarified that we are in the effective range by adding the following sentences: “2DG is typically used in the range of 5-10 mM in cell culture studies and in general, has anti-proliferative effects. To test whether 2DG treatment was in the effective range, explants were exposed to BrdU, which is incorporated into S-phase cells, for 30 minutes prior to harvesting. 2DG treatment resulted in a dose-dependent inhibition of RPC proliferation as evidenced by a reduction in BrdU^+^ cells (Figure 3D), indicating that our treatment was in the effective range.” (lines 246-251).

(6) Figure 3F: The authors use immunostaining for cleaved, activated caspase-3 to assess the amount of apoptotic cell death. However, there are many different possible mechanisms for neuronal cells to die, the majority of which are caspase-independent. To assess the amount of cell death occurring, the authors should perform a TUNEL assay (which labels apoptotic and non-apoptotic forms of cell death; Grasl-Kraupp et al., Hepatology 21:1465-8, 1995), quantify the numbers of TUNEL-positive cells in the retina, and compare this to the numbers of cells positive for activated caspase-3.

We agree with the reviewer that there are more ways for a cell to die than just apoptosis, and TUNEL would pick up dying cells that may undergo apoptosis or necrosis, for example, our data with cleaved caspase-3, an executioner protease for apoptosis, provides us with clear evidence of cell death in our different conditions. Since this manuscript is not focused on cell death pathways, we have not performed the additional TUNEL assay.

(7) Figure 4F and 4I: At post-natal day P7 the rod outer segments (OSs) only just start to grow out and the characteristic, rhodopsin-filled disk stacks are not yet formed. To test whether the PFKFB3 gain-of function or the Pten KO has a marked effect on OS formation and length, the authors should perform the same tests on older, more mature retina at a time when rod OS show their characteristic disk structures (e.g. somewhere between P14 to P30). The same applies to the 2DG inhibition on the Pten KO retina.

The precocious differentiation of rod outer segments observed in P7 *Pten*-cKO retinas does not persist in adulthood, and instead reflects a developmental acceleration. Indeed, we found that in *Pten* cKO retinas at 3-, 6- and 12-months of age, rod and cone photoreceptors degenerate, and cone outer segments are shorter (Hanna et al., 2025; Tachibana et al., 2016). These data demonstrate that *Pten* is required to support rod and cone survival.

(8) Figure 5: Lowering media pH is a rather coarse and untargeted intervention that will have multiple metabolic consequences independent of PKM2. It is thus hardly possible to attribute the effects of pH manipulation to any specific enzyme. To assess this and possibly confirm the results obtained with low pH, the authors should perform a targeted inhibition experiment, for instance using Shikonin (Chen et al., Oncogene 30:4297-306, 2011), to selectively inhibit PKM2. If the retinal explant cultures contained the RPE, an additional question would be how the changes in RPE would alter lactate flux and metabolization between RPE and neuroretina (see also question 4 above).

We have reframed the rationale for the pH manipulation experiments, highlighting the importance of pH in cell fate specification, and indicating that the aggregation of PKM2 is only one possible effect of lower pH.

We wrote: “Given that altered glycolysis influences intracellular pH, which in turn controls cell fate decisions, we set out to assess the impact of manipulating pH on cell fate selection in the retina. One of the expected impacts of lowering pH was the aggregation of PKM2, a rate-limiting enzyme for glycolysis, which aggregates in reversible, inactive amyloids (Cereghetti et al., 2024).” (lines 362-366).

We have also added a discussion point “Whether pH manipulations also impact the stability of other retinal proteins, such as PKM2, can be further investigated in the future using specific PKM2 inhibitors, such as Shikonin (Chen et al., 2011). (lines 545-547).

(9) Figure 5G: As for Figure 3F, the authors should perform TUNEL assays to assess the number of cells dying independent of caspase-3.

Please see response to point 6.

(10) Figure 7E: In the figure legend "K" should read "E". From the figure and the legend, it is not clear to which cell type this diagram should refer. This must be specified. Importantly, the insulin-dependent glucose-transporter 4 (GLUT4) highlighted in Figure 7E, while expressed on inner retinal vasculature endothelial cells, is not expressed in retinal neurons. What GLUTs exactly are expressed in what retinal neurons may still be to some extent contentious (cf. Chen et al., elife, https://doi.org/10.7554/eLife.91141.3 ; and reviewer comments therein), yet RPE cells clearly express GLUT1, photoreceptors likely express GLUT3, Müller glia cells may express GLUT1, while horizontal cells likely express GLUT2 (Yang et al., J Neurochem. 160:283-296, 2022).’

We have removed this summary schematic for simplicity.

(11) Materials and methods: The retinal explant culture system must be described in more detail. Important questions concern the use of medium and serum for which the providers, order numbers, and batch/lot numbers (whichever is applicable) must be given. The glucose concentration in the medium (including the serum content) should be measured. A key concern is whether the explants were cultivated submerged into the medium - this would prevent sufficient oxygenation and drive metabolism towards glycolysis (i.e. the Pasteur effect) - or whether they were cultivated on top of the liquid medium, at the interface between air and liquid (i.e. a situation that would favor OXPHOS).

We have added further detail to the methods section for the explant assay (lines 686-689). We cultured the retinal explants on membranes on top of the media, which is the standard methodology in the field and in our laboratory (Cantrup et al., 2012; Tachibana et al., 2016; Touahri et al., 2024). Typically, RPCs undergo aerobic glycolysis, meaning that even in the presence of oxygen, they still prefer glycolysis rather than OXPHOS. We demonstrated that 2DG blocks RPC proliferation when treated with 2DG, indicating that RPCs are indeed favoring glycolysis in our assay system.

(12) A point the authors may want to discuss additionally is the potential relevance of their data for the pathogenesis of human diseases, especially early developmental defects such as they occur in oxygen-induced retinopathy of prematurity.

We would like to thank the reviewer for their valuable comment. Given that retinopathy of prematurity (ROP) is primarily vascular in nature, and we have not investigated vascular defects in this study, we have elected not to add a discussion of ROP to our manuscript.

Minor points(1) Please add a label indicating the ages of the retina to images showing the entire retina (i.e. "P7"; e.g. in Figures 1F, 3, 4D, 5, etc.).

Figure 1:

1D: E18.5 indicated at the bottom of the two panels

1F – P0 is indicated at the bottom of the two panels.

Figure 3C-H: P0 explant stage and days of culture indicated

Figure 4D: E12.5 BrdU and P7 harvest date indicated

Figure 5C-H: P0 explant stage and days of culture indicated

Figure 7A-E: P0 explant stage and days of culture indicated

(2) The term Ctnnb1 should be introduced also in the abstract.

We now state that *Ctnnb1* encodes for b-catenin in the abstract.

(3) Line 249: "...remaining..." should probably read "...remained...".

Changed (now line 260).

(4) Line 381: The sentence "...correlating with the propensity of some RPCs to continue to proliferate while others to differentiate.", should probably be rewritten to something like "...correlating with the propensity of some RPCs to continue to proliferate while others differentiate.".

We have corrected this sentence.

(5) The structure of the discussion might benefit from the introduction of subheadings.

We have introduced subheadings.

**Reviewer #3 (Recommendations for the authors):**
(1) Figure 1H shows the kinetics of rod photoreceptor production as accelerated, but does not represent the fact that fewer rods are ultimately produced, which appears to be the case from the data. If so, the Pten cKO curve should probably be lower than WT to reflect that difference.

We have removed this graph (as per Reviewer #2, point 2).

(2) KEGG analysis also showed that the HIF-1 signaling pathway is altered in the Pten cKO retina. What is the significance of that, and is it related to metabolic dysregulation? It has been shown that lactate can promote vessel growth, which initiates at birth in the mouse retina.

We have added some information on HIF-1 to the Discussion. “The increased glycolytic gene expression in *Pten*-cKO retinas is likely tied to the increased expression of hypoxia-induced-factor-1-alpha (*Hif1a*), a known target of mTOR signaling that transcriptionally activates *Slc1a3* (GLUT1) and glycolytic genes (Hanna et al., 2022). Indeed, mTOR signaling is hyperactive in *Pten*-cKO retinas (Cantrup et al., 2012; Tachibana et al., 2016; Tachibana et al., 2018; Touahri et al., 2024), and likewise, in *Tsc1-c*KO retinas, which also increase glycolysis via HIF-1A (Lim et al., 2021).” (lines 489-494).

Cantrup, R., Dixit, R., Palmesino, E., Bonfield, S., Shaker, T., Tachibana, N., Zinyk, D., Dalesman, S., Yamakawa, K., Stell, W. K., Wong, R. O., Reese, B. E., Kania, A., Sauve, Y., & Schuurmans, C. (2012). Cell-type specific roles for PTEN in establishing a functional retinal architecture. PLoS One, 7(3), e32795. https://doi.org/10.1371/journal.pone.0032795

Cereghetti, G., Kissling, V. M., Koch, L. M., Arm, A., Schmidt, C. C., Thüringer, Y., Zamboni, N., Afanasyev, P., Linsenmeier, M., Eichmann, C., Kroschwald, S., Zhou, J., Cao, Y., Pfizenmaier, D. M., Wiegand, T., Cadalbert, R., Gupta, G., Boehringer, D., Knowles, T. P. J., Mezzenga, R., Arosio, P., Riek, R., & Peter, M. (2024). An evolutionarily conserved mechanism controls reversible amyloids of pyruvate kinase via pH-sensing regions. Dev Cell. https://doi.org/10.1016/j.devcel.2024.04.018

Chen, J., Xie, J., Jiang, Z., Wang, B., Wang, Y., & Hu, X. (2011). Shikonin and its analogs inhibit cancer cell glycolysis by targeting tumor pyruvate kinase-M2. Oncogene, 30(42), 4297-4306. https://doi.org/10.1038/onc.2011.137

Hanna, J., Touahri, Y., Pak, A., David, L. A., van Oosten, E., Dixit, R., Vecchio, L. M., Mehta, D. N., Minamisono, R., Aubert, I., & Schuurmans, C. (2025). Pten Loss Triggers Progressive Photoreceptor Degeneration in an mTORC1-Independent Manner. Invest Ophthalmol Vis Sci, 66(3), 45. https://doi.org/10.1167/iovs.66.3.45

Tachibana, N., Cantrup, R., Dixit, R., Touahri, Y., Kaushik, G., Zinyk, D., Daftarian, N., Biernaskie, J., McFarlane, S., & Schuurmans, C. (2016). Pten Regulates Retinal Amacrine Cell Number by Modulating Akt, Tgfbeta, and Erk Signaling. J Neurosci, 36(36), 9454-9471. https://doi.org/10.1523/JNEUROSCI.0936-16.2016

Touahri, Y., Hanna, J., Tachibana, N., Okawa, S., Liu, H., David, L. A., Olender, T., Vasan, L., Pak, A., Mehta, D. N., Chinchalongporn, V., Balakrishnan, A., Cantrup, R., Dixit, R., Mattar, P., Saleh, F., Ilnytskyy, Y., Murshed, M., Mains, P. E., Kovalchuk, I., Lefebvre, J. L., Leong, H. S., Cayouette, M., Wang, C., Sol, A. D., Brand, M., Reese, B. E., & Schuurmans, C. (2024). Pten regulates endocytic trafficking of cell adhesion and Wnt signaling molecules to pattern the retina. Cell Rep, 43(4), 114005. https://doi.org/10.1016/j.celrep.2024.114005